# A Novel Fermented Rapeseed Meal, Inoculated with Selected Protease-Assisting Screened *B. subtilis* YY-4 and *L. plantarum 6026*, Showed High Availability and Strong Antioxidant and Immunomodulation Potential Capacity

**DOI:** 10.3390/foods11142118

**Published:** 2022-07-16

**Authors:** Yan Wang, Hao Sun, Xiaolan Liu

**Affiliations:** College of Food and Biological Engineering, Qiqihar University, Qiqihar 161006, China; s18814523213@163.com (H.S.); wangyan3522186@126.com (X.L.)

**Keywords:** rapeseed meal, FRSMP, molecular weight, antioxidant activity, availability, immunomodulation activity

## Abstract

A study was conducted to investigate the yield of small peptides from rapeseed meal (RSM) by solid-state fermentation (SSF) with acid-protease-assisting *B. subtilis* YY-4 and *L. plantarum CICC6026* (FRSMP). This study explored the availability, antioxidant capacity and immunomodulation activity. The objective of this study was to develop a novel functional food ingredient to contribute to health improvement. The results showed that the concentrations of soluble peptides and free amino acids significantly increased after fermentation (*p* < 0.001), and the concentration of small molecular peptides (molecular weight < 1 KDa) significantly increased (*p* < 0.001). The dense surface microstructure of the RSM after fermentation was changed to be loose and porous. The FRSMP exhibited high availability and high antioxidant activity, and it displayed high immunomodulation activity. The novel fermentation was effective for improving the nutritional and biological properties, which provided a feasible method of enhancing the added value.

## 1. Introduction

Rapeseed (Figure 1) is the third-most abundant oil crop worldwide, and its by-product (rapeseed meal, RSM) not only has high annual output (more than 14 million in China), but also has high crude protein content (30–42%) and well-balanced amino acid compositions with high quantities of indispensable amino acids, especially sulfur-containing amino acids [1]. Although RSM has many advantages, it is usually used as feed ingredients due to the presence of anti-nutritional factors (ANFs) (such as glucosinolate, tannin, phytic acid and complex fiber) and macro-molecule proteins [2], which limits the usage of the rapeseed protein in food applications (Laguna et al. 2019; Jia, 2021), thereby resulting in a low added value. Many studies demonstrate that the hydrolysates of rapeseed polypeptides display high utilization, and also present high antioxidant activity [3,4]. The rapeseed peptide (Pro-Ala-Gly-Pro-Phe) from rapeseed protein hydrolysate has a high DPPH scavenging rate [5]. The rapeseed peptides hydrolyzed by flavourzyme and alcalase show a high reducing antioxidant power and hydroxyl radical scavenging ability and exhibit highly inhibited activity, with 50% in the serum MDA level [6]. However, the rapeseed peptides are inactive within parent proteins, and their specific bioactivities are based on amino acid compositions and sequences [7].

The bioactivity of rapeseed has been a hotspot of food research in recent years, particularly focusing on the explorations of byproducts with natural sources of bioactive compounds [8]. Protease, with high selectivity, is a preferred tool for obtaining hydrolysates with specific peptides, but the presence of ANFs can decrease the protease activity, which limits the release of highly active rapeseed peptides. Microbial fermentation is observed as an efficient, economical technique, and it is usually applied to detoxify and improve nutritional values. Meanwhile, bioactive compounds with health benefits might be formatted during fermentation, especially small peptides and free amino acids (FAAs), which are considered to be the most important products [4]. In addition, the growing interest in health-promoting whole natural foods has contributed to the current resurgence of fermented foods [9].

Traditional fermentation is pure culture fermentation, and few metabolites are produced, resulting in single function and flavor. Tangyu, Muller, Bolten and Wittmann [10] report that mixed microorganisms with two or more microbial species promisingly produce synergistic effects to improve physiological functions, which is due to the various enzymes produced by multi-microbial species to remove the ANFs and increase the hydrolytic sites, thereby facilitating microbial protease to break down macro-molecular proteins into small molecular peptides and FAAs. The small molecular peptide content is positively related to the bioactivity. Adeola et al. [11] demonstrate that bioactive peptides are usually composed of 2–10 amino acids, and small peptides with low molecular weight (less than 1000 Da) display high antioxidant activity. The exogenous protease may cooperate with microorganisms to enrich the protease system during fermentation, thereby increasing the release of bioactive peptides. Cheng et al. [12] demonstrate that the effects of *B**. subtilis* and enzymes on immune response are better than those of *B**. subtilis* alone.

The selection of microbial strains is very critical for the final products of fermentation, and the selected microorganisms must be high-producing strains of rapeseed peptides, as well as those that release more peptides with highlighted physiological activities. Strains of *Bacillus* are considered good producers of active peptides [13]. Apart from *Bacillus*, *Lactobacillus* is a source of bioactive peptides and phenols [14]. *B. subtilis* is a dominant culture that secretes large quantities of extracellular enzymes, and the proteolysis system of *Lactobacillus* combines the actions of proteinases and peptidases and is particularly used in debittering starter adjuncts. RSM fermented with *L. planrum, B. licheniformis* and *S. utilis* can degrade the glucosinolate and increase the peptide content [15], and RSM fermented by *B**. subtilis* presented high antioxidant content [16]. The soy products fermented by *Lactobacillus* increased the total amount of antioxidant ingredients [17].

Many studies demonstrate that antioxidant peptides can increase immune capacity. Jiang et al. [18] reported that antioxidant peptides (corn peptides fermented with LAB) had immune functions for calves during the postweaning stage, and Rezazadeh, Kowsar, Rafiee and Riasi [19] found that the soybean meal fermented with *B**. subtilis* GR-101 improved the immune functions of weaned calves. Many studies have demonstrated the antioxidant activity of rapeseed peptides [15,16], but few studies have examined the effects of rapeseed peptides on immunomodulation capacity.

Few studies have involved increasing rapeseed peptide yield by the synergistic effects of screening strains and exogenous protease, and efforts have been made to develop a more efficient method to prepare more acceptable products for the benefit of humans. A novel fermentation of rapeseed meal with selected protease-assisting screened strains (FRSMP) was developed. To obtain more peptides, the screened strain was employed, and an optimized strain combination was selected. In addition, a comprehensive method (Plackett–Burman and Box Behnken design) was also developed to optimize the fermentation parameters, thereby maximally increasing the release of rapeseed peptides. Furthermore, the availability and biological properties were investigated, which provides an effective method to develop a functional food ingredient that contributes to health.

## 2. Materials and Methods

### 2.1. Materials

*L. plantarum* (CICC6026) and *B. natto* (CICC10262) were purchased by CICC, *B. subtilis* (1.0504), *B. licheniformis* (1.0813), *B. coagulans* (1.2009) and *L. buigaricus* (1.148) were purchased by CGMCC, and *L. acidophilus* (ACCC10637) and *B. subtilis* (ACCC10619) were purchased by ACCC. RSM was collected from three regions (Shandong, Sichuan and Zhejiang) in China, and were ground through 0.4–0.5 mm sieves in a Wiley mill. Acid protease (50,000 U/g; pH: 2.5–6.0) was purchased from Doing-Higher Bio-Tech Co., Ltd. (Nanning, China), and neutral protease (50,000 U/g, pH: 5.5–7.5), alkaline protease (50,000 U/g, pH: 8.0–10.0) and flavor protease (50,000 U/g, pH: 6.0–9.0) were purchased from Doing higher Co., Ltd. The reagents were purchased from Sigma (St. Louis, MO, USA).

### 2.2. Method

#### 2.2.1. Preparation of the FRSMP

The strain from the soil was primarily screened by measuring the capacity of degrading glucosinolate (extracted glucosinolate from the rapeseed meal as the sole carbon source) according to the method described by Albaser et al. [20] with slight modification, and was secondarily screened by measuring the capacity of hydrolyzing protein (rapeseed protein as the sole protein source), and the strain was identified as *B. subtilis* by 16 rRNA sequence in Shanghai Sangon Bioengineering Co., Ltd (Shanghai, China). Then, the strain *B. subtilis YY-4* was mutated and bred by UV and chemistry methods to obtain a higher capacity for producing protease. The co-fermented strain combination was subsequently screened by measuring the capacity of hydrolyzing rapeseed protein, and the optimal protease assisting the best combination of strains was screened by the same method. Based on the results of earlier single-factor laboratory experiments (acid protease amount, moisture, inoculation size, time, temperature and sugar), the screening of the most significant fermentation parameters affecting peptide yield was performed using a Plackett–Burman design (PBD). Subsequently, a response surface methodology (RSM) design was used to optimize for enhancing the peptide yield. The 6 variables of PBD were evaluated at 2 levels (protease amount: 1, 6 mg/g; temperature: 32, 38 °C; inoculation size: 2, 8 mL/g; time: 3, 6 d; sugar amount: 0, 6 mg/g; moisture: 400, 600 mL/kg).

The significant variables (protease amount, moisture and time) were obtained according to the results obtained from the PBD, and they were used to design the RSM experiment for enhancing the peptide yield at 3 levels, and the optimized values were validated by repeating the experiment in three flasks. These 3 variables selected for optimization were coded as A (time), B (acid protease amount) and C (moisture) with three levels (−1, 0, and 1) per variable, and the variables and their value ranges were put into the BBD. The BBD proposed 17 sets of conditions for further experimentation, and the optimized condition obtained from BBD was validated by conducting the experiments (3 times for each fermentation experiment).

#### 2.2.2. Nutritional Compositions and Microstructure of the RSM before and after Fermentation

The crude protein (CP) concentration was analyzed according to AOAC procedure [21]. The concentration of neutral detergent fiber (NDF) was determined using the Ankom system (Ankom 220 fiber analyzer; ANKOM Technology Corp, Macedon, NY, USA). The concentration of glucosinolate was determined by the anthrone colorimetry [22]. The concentration of soluble peptides was determined by the Folin phenol method [23]. The lactic acid content was determined by gas chromatography [24]. The determination of total phenolic content was based on the Folin–Denis method, and was conducted using a slightly modified method described by Laguna [25]. The phytic acid content was determined using the method described by Wheeler and Ferrel [26]. The surface morphology and microstructure were observed with a high-resolution field emission scanning electron microscope system using the secondary electron detector (SEM, Hitachi S-4300) (acceleration voltage, 5.0 kV; magnification, 5000×). The different fractions of the RPs were separated and collected by the ultrafiltration method. The molecular cut-off weights (MWCO) of the membranes used sequentially were 1, 3, 5 and 10 kDa, respectively. The material of the ultrafiltration membrane was Xampler laboratory-scale hollow fiber cartridges, and the styles of membrane (GE Healthcare Bio-Sciences Corp. Westborough, MA 01581, USA) were UFP-1-C-3M, UFP-3-C-4×2MA, UFP-5-C-4×2MA and UFP-10-C-4×2MA. The QuixStand system (QSM-03S, 564107-42) was equipped with a 1 L liter QixStand reservoir with a cap and downcomers without pumps, and was equipped with a pump (Watson-Marlow Bredel Pump, Falmouth Cornwall TR11 4RU England, SN: 0122406). The RPs were ultrafiltrated at a trans-membrane pressure (TMP) of 150 kPa (45 °C) and at a volume reduction factor (VRF) of ½, and the selection of the condition relied on the SDS pattern of the retentate compartment. The specific procedure of the ultrafiltration was shown in Figure 2.

#### 2.2.3. Digestibility of the DM and CP of the RSM before and after Fermentation

The digestibility of the RSM and FRSMP was determined by the simulated digestive system of monogastric animals (SDS-II) created by the Chinese Academy of Agricultural Sciences, and the control software was Version 1.0 (Ruan Zhu Deng Zi No. 0154149), which was designed according to the principle of Wilfart et al. [27]. In principle, the SDS-II system simulates the digestive tract as a simulated digestive system of monogastric animals in vitro, wherein the simulation is realistic and repeatable. The details were as follows: the dialysis bag (MEMBRA-CEL MD44-14, Viskase, Lombard, IL, USA) was boiled for 10 min at 2% NaHCO_3_ (W/V), and then boiled for 10 min at 1 mmol/L EDTA (pH 8.0) and was stored at 4 °C. The 1 g of rapeseed meal was put into a dialysis bag (patent number: ZL 2009 2 0105937.X), and then was put into the simulated digest liquid. It was digested for 4 h in 20 mL of simulated gastric juice (hydrochloric acid solution as buffer, pH 2.0), and then was digested for 7.5 h in 2 mL of simulated small intestine fluid (buffer pH, 6.5) (at the early stage) and then digested again for 7.5 h in the same fluid (buffer pH, 7.9) (at the late stage). The water bath system was set at 43 °C and 180 r/min, and the flow rate of the peristaltic pump was 425 mL/min. The preparation methods of gastric juice and intestinal juice were in accordance with the test method of Zheng [28].

#### 2.2.4. Molecular Weight Distribution and Free Amino Acid Profile of the RSM before and after Fermentation

The molecular weight distributions of soluble peptides were preliminarily determined by SDS-PAGE with the protein markers (lowest, 9.5 kDa), and the FERSM were further analyzed using a high-performance liquid chromatograph equipped with a Superdex peptide PE/7.5/300 (ID 7.5 × 300 mm) and a UV detector. The approtinine (MW, 6500 Da), blue dextran (MW, 2000 Da), bacitraein (MW, 1400 Da), glutathione disulfide (MW, 610 Da) and reduced glutathione (MW, 310 Da) were used to create a calibration curve of MW relative to retention time for calculating the MWs of peptide fractions. The chromatographic conditions are as follows: flow rate, 0.25 mL/min; mobile phase, acetonitrile/water/trifluoroacetic acid (45/55/0.1, *v/v/v*); column temperature, 30 °C; detection wavelength, 214 nm; standard protein load amount, 100 uL; peptide concentration, 2 mg/mL. The concentrations of the FAAs were analyzed with HPLC-MS/MS (shimadzu LC20AD-API 3200MD TRAP), and the amino acid kit (MSLAB-45+AA) was provided by Beijing Mass Spectrometry Medical Research Co., Ltd. (batch number, MSLAB451561)(Beijing, China).

#### 2.2.5. Evaluation of the Antioxidant Activity of the RSM before and after Fermentation In Vitro

The samples from FERSM were ground and passed through a 60-mesh sieve and extracted with 1.00 mL of 0.03 mol/L Tris-HCl buffer (pH 8.0) for 2 h, then were centrifuged at 10,000× *g* for 10 min (4 °C) to obtain the RPs. The RPs were isolated and purified according to the method described by Sun et al. [29] and Lin et al. [30] with some modifications. The mineral salts in the filtrate were removed using a TORAY nanofiltration membrane (MW cut-off 150 Da, General Electric Company, USA)(Boston, USA), and the membrane was installed in multifunctional separation equipment (HYM-Multi-RN with Wanner/g13, Haoyuan Membrane Technology Co., Ltd., Xiamen, China). The purification method followed the method described by Jiang, Cui, Wang, Xu and Zhang [31].

The peptides were used for the analysis of antioxidant capacity. The antioxidant activities were measured using the following tests. The 2,2-Diphenyl-1-picrylhydrazyl (DPPH) scavenging activity, the ferrous-ion-chelating activity and the reducing power were measured with the methods from Zhang, Wang, Xu and Gao [5]. The effective concentration (EC_50_) that was required to achieve 50% activity was adopted according to the method of Xiao et al. [32], and the EC_50_ values were obtained by nonlinear regression analysis.

#### 2.2.6. The Availability and Bioactivity In Vivo of the RSM before and after Fermentation

The 90 weaned male rats weighing 60–70 g were purchased from Beijing, and the rats were randomly allocated to two dietary groups (one control group and one experiment group), and each treatment group contained 3 replicates of 15 rats. The diet compositions and nutrient ingredients are shown in Table 1, and the diet of the rats is formulated according to the GB14924.3-2001. The rats were fed once daily for 21 consecutive days and were acclimatized for 3 days prior to the experiment. The rats were maintained in a controlled environment (temperature 22–25 °C, humidity 50–70%), and were housed in stainless steel cages lined with husk and water *ad libitum.*

Animal experiments were conducted according to the principles and guidelines approved by the Animal Care and Use Committee of Northeast Agricultural University. The average daily weight gain (ADG) and dry matter feed intake (DMI) of the rats were measured by weighing each day, and the feed conversion rate (FCR) was calculated. At the end of the experiment, the rats were fasted for 12 h, and the eyeball blood and the jejunum tissues were collected and analyzed.

The levels of IgG, IgA, IgM, triiodothyronine (T_3_), thyronine (T_4_), insulin-like growth factor1 (IGF-1), growth hormone (GH), interleukin-1β (IL-1β), interleukin-6 (IL-6) and secretory immunoglobulin A (SigA) in serum were measured via ELISA kits from Beijing Sino-UK Institute of Biological Technology (Huawei Delang DR-200BS enzyme label analyzer), and all procedures were performed according to the manufacturer’s instructions. The jejunum tissues of the rats were homogenized in PBS containing protease inhibitors and centrifuged at 10,000× *g* and 4 °C for 10 min, then the supernatant was collected and analyzed for the tissue IL-1β, IL-6 and SigA levels.

#### 2.2.7. Statistical Analysis

All experiments were performed in triplicate. Minitab 17.0 software (Plackett–Burman design) was used to identify the fermentation parameters that had significant effects on RSMP yield and to generate Pareto charts. The Design Expert ^®^ version 12.0 (Stat-ease, Minneapolis, MN, USA) (response surface methodology) was used to obtain the predicted model and optimal fermentation condition. The nutrient concentrations of the RSM and FRSMP were compared using the GLM procedure of the SAS 9.13 [33], and significant differences among treatments were assessed with Duncan’s new multiple range test. Differences were considered statistically significant at *p* < 0.05. The animal experimental data were analyzed as a randomized complete block design by ANOVA using SAS 9.13, and the following model was adopted: Y ij = u + Ti + Ei, where Y ij = dependent variable, u = the overall mean, Ti = treatment effect, Ei = error term.

## 3. Results and Analyses

### 3.1. Establishment of Novel Fermentation System

#### 3.1.1. Screening of Strains and Exogenous Protease

The data of screening strains and strain combinations are presented in Figure 3a,b. The soluble peptide yield of rapeseed meal fermented with *B. subtilis* YY-4 was higher than those of purchased strains (*p* < 0.05), and the combination (*B. subtilis* YY-4 and *L. plantarum* 6026) had higher peptide yield than other combinations (*p* < 0.05). The data of selected proteases are presented in Figure 3c, and the peptide yield of acid protease assisting the best combination increased more than those of other proteases (*p* < 0.05).

#### 3.1.2. Optimization of Fermentation Condition

The influence of fermentation variables on soluble peptide yield is presented in the Pareto Chart (Figure 4a) (PBD). The Pareto Chart shows that the parameters with a significant effect (*p* < 0.05) were acid protease amount (*p* = 0.000), moisture (*p* = 0.010) and time (*p* = 0.001), and these three variables should be further optimized using a BBD design. The linear regression equation was derived as follows: Y = 174.49 + 3.73 Temperature + 22.59 Acid protease amount + 0.51 Ammonium sulfate + 4.08 Inoculation size + 3.06 Sugar + 14.76 Time–6.94 Moisture (*p* < 0.0001, R^2^ = 0.99), which indicated that the model was sufficient for predicting peptide yield.

The design and ANOVA results of the quadratic regression model (RSM) optimized by BBD are given in Figure 4b,c. The F and *p* values (77.85 and *p* < 0.0001, respectively) proved that the model was highly significant, and the results suggested that the selection of the variable levels was reasonable and the maximum response value was likely to be predicted within the selected range. The maximum response value was obtained in run 8 (227 mg/g), followed by run 12, 15 and 1 (225, 225 and 224 mg/g, respectively), which indicated that the middle levels of the variables had a higher yield. The model predicted a maximum of 230.56 mg/g, and the optimal conditions were as follows: acid protease, 0.53 mg/g; moisture, 46%; inoculation size, 4.78 mL/100 mg; time, 5 d; temperature, 37 °C; sugar, 3 mg/g. The quadratic polynomial equation derived from regression was as follows: Y = 224.2 + 14.25 A + 15 B + 7.25 C + 7 AB + 3 AC + 10.50 BC − 19.85 A^2^ − 33.35 B^2^ − 22.85 C^2^, where A, B, C were coded values of fermentation time, acid protease amount and moisture, respectively. The model was highly significant (*p* < 0.0001), and the coefficient of determination was 0.99, which suggested that the model could be used to predict the optimum condition. The least fit was also used to judge no significance (*p* = 0.0643), which further indicated that the predicted model was adequate. According to the F values, the influence of the three variables on peptide yield was B > A > C.

The 3D plots in response surface methodology are shown in Figure 4d–f. It was predicted that the highest peptide yield was acquired under the range of designed conditions. To further confirm the adequacy of the predicted values and the model, an additional fermentation experiment was performed under the optimal fermentation conditions. The actual average peptide yield was 231.05 ± 5.6 mg/g, which was very close to the predicted value, indicating that the suggested model was valid. 

### 3.2. Surface Morphology and Microstructure of the RSM before and after Fermentation

The surface morphology and microstructure of the RSM before and after fermentation are presented in Figure 5. The dense surface microstructure of the RSM after fermentation was changed to be loose and porous, and the crystalline and amorphous regions of the fiber were destroyed, which increased the contact area between protease and rapeseed proteins, thereby improving the efficiencies of microbial protease and acid protease.

### 3.3. Chemical Compositions of the RSM before and after Fermentation

The chemical compositions of the RSM and FRSMP are presented in Table 2. The glucosinolate and phytic acid concentrations of the FRSMP substantially decreased compared to the RSM (*p* < 0.001), and the NDF concentration of the FRSMP decreased to 140 mg/g (*p* < 0.001). The CP concentration of the FRSMP increased to 405 mg/g compared to the RSM (*p* = 0.017), and the soluble peptide concentration of the RSM after fermentation significantly increased from 53.33 mg/g to 231.05 mg/g (*p* < 0.001). The FRSMP had no differences in total phenolic concentration compared to the RSM (*p* > 0.05), and it had a significant increase in lactic acid concentration (*p* < 0.001).

### 3.4. Molecular Weight Distribution and Free Amino Acid Profile of the RSM before and after Fermentation

The molecular weight distributions of soluble peptides determined by SDS-PAGE of the RSM before and after fermentation are presented in Figure 6a, and the concentrations of different peptide fractions are shown in Table 2. The molecular weight was a critical variable responsible for the activity of peptides. Most rapeseed proteins with high molecular weight were degraded to small rapeseed peptides with low molecular weight after fermentation, and the molecular weights of most rapeseed proteins were degraded to below 9.5 kDa (Y = −6.3013X + 33.134, R^2^ = 0.9602) (SDS-PAGE).

To further determine the MW distribution of small molecular peptides accurately, the gel chromatography method was employed. The calibration curve of MW relative to retention time is shown in Figure 6b, and the MW distribution of the RSM and FRSMP is shown in Figure 6c,d. The MW distribution was determined according to a calibration curve of MW relative to retention time. The small peptide concentrations of RSM obviously increased after fermentation, and the peak areas of the FRSMP with low MW were more than that of the RSM. This result suggested that the RSM processed with acid-protease-assisting *B. subtilis* YY-4 and *L. plantarum* CICC6026 (FRSMP) was an effective way to produce low molecular weight oligopeptides. Compared to macro-molecular proteins, small peptides can not only be directly absorbed by animal intestines, but also exhibit immunomodulation activity, antioxidant activity and so on.

The concentrations of the FAAs of the RSM and FRSMP are presented in Table 3. The total concentration of the FAAs of the FRSMP significantly increased (*p* < 0.001) compared to that of the RSM, ranging from 3630.80 to 7914.31 ng/g DM. After fermentation, the total concentration of the essential amino acids (EAAs) had a significant increase (*p* < 0.001), and the proportion of the EAAs to NEAA markedly increased, ranging from 0.29 to 0.97. For all the amino acids, the concentration of the second restrictive amino acid (methionine) of the FRSMP changed the most, and was 121.81 times higher than that of the RSM. The concentrations of the glutamate (2.89-fold, *p* < 0.001) and lysine (8.18-fold, *p* < 0.001) significantly increased after fermentation, which was responsible for the delicious taste of products. The concentrations of the valine (2.14-fold, *p* < 0.001), cysteine (2.82-fold, *p* < 0.001), leucine (5.91-fold, *p* < 0.001), phenylalanine (6.32-fold, *p* < 0.001) and tryptophan (1.99-fold, *p* < 0.001) significantly increased after fermentation, and these amino acids were usually considered to have antioxidant activities.

### 3.5. Availability In Vitro and In Vivo of the RSM before and after Fermentation

The effects of the RSM and FRSMP on availability in vitro and in vivo of weaned rats in vivo are presented in Table 4. The DM and CP digestibility in vitro of the RSM and FRSMP are presented in Table 4. The DM digestibility of the FRSMP was significantly higher than that of the RSM (*p* < 0.01), and the CP digestibility of the FRSMP significantly increased compared to that of the RSM (*p* = 0.012). The ADG values of weaned rats fed FRSMP significantly increased compared to those fed RSM (*p* < 0.05), and the DMI and FCR value had no significant increase (*p* = 0.088 and *p* = 0.115). The serum IGF-1 levels of weaned rats fed FRSMP significantly increased compared to those fed RSM (*p* = 0.011), and the tissue IGF-1 levels significantly increased (*p* < 0.001). The weaned rats fed FRSMP had higher T_3_ and T_4_ levels in serum than those fed RSM (*p* < 0.05), and had an increased tendency for the serum GH level, but had no significant difference (*p* > 0.05).

### 3.6. Antioxidant Activity In Vitro of the RSM before and after Fermentation

#### 3.6.1. DPPH Scavenging Activity

The DPPH scavenging activities of the RSM and FRSMP are presented in Figure 7a. The DPPH radical scavenging activities of the RSM and FRSMP increased steadily in the concentration range (0–1 mg/mL), and achieved maximum equilibrium value at a concentration of 900 µg/mL. The DPPH scavenging activity of the RSM reached 51.69% at a concentration of 900 µg/mL, while that of the FRSMP reached 81.66% at a same concentration. The EC_50_ values of DPPH radical scavenging activities of RSM and FRSMP were estimated to be 0.48 mg/mL and 0.79 mg/mL, respectively, and the EC_50_ value was inversely related with the antioxidant activity, which suggested that the RSM after fermentation had a high DPPH scavenging activity.

#### 3.6.2. Ferrous-Ion-Chelating Activity

The ferrous-ion-chelating activities of the RSM and FRSMP are presented in Figure 7b. The ferrous-ion-chelating activity was positively correlated to the concentration, and it reached 75.99% when the concentration of the rapeseed peptides from the FRSMP was 6 mg/mL. The result was higher than the value reported by Rong et al. [16], who reported that the ferrous-ion-chelating activity of the fermented RSM was 60% at the same concentration. The ferrous-ion-chelating activity of the RSM at a concentration of 6 mg/mL was the same as that of the FRSMP at a concentration of 3.25 mg/mL, and the EC_50_ values of the RSM and FRSMP were estimated to be 4.65 mg/mL and 2.75 mg/mL, respectively, which indicated that the FRSMP displayed a higher ferrous-ion-chelating activity compared to the RSM.

#### 3.6.3. Reducing Power

The reducing powers of the RSM and FRSMP are presented in Figure 7c. The absorbance values of the reducing power of the RSM and FRSMP depended on concentrations and were enhanced with the corresponding increases in the concentrations. When the absorbance value was 1, the concentrations of the rapeseed peptides from the RSM and FRSMP were 0.64 mg/mL and 0.85 mg/mL, respectively, which indicated that the FRSMP displayed a higher reducing power compared to the RSM.

### 3.7. Immunomodulation Activity In Vivo of the RSM before and after Fermentation

The effects of the RSM and FRSMP on immunomodulation activity of weaned rats are presented in Table 5. The serum IgM and IgG levels of weaned rats fed FRSMP were greater than those fed RSM *(p* < 0.05), and the tissue SigA levels significantly increased (*p* = 0.039). Weaned rats fed FRSMP had lower IL-6 levels in serum and tissue than those fed RSM (*p* < 0.001).

## 4. Discussion 

### 4.1. Discussion on the Novel Fermentation System

#### 4.1.1. Discussion on the Novel Fermentation Method

The capacity of hydrolyzing rapeseed protein of *B. subtilis* YY-4 was higher than those of others, which was due to the strain screened measuring the hydrolyzing rapeseed protein capacity. The rapeseed peptide yield of *B. subtilis YY-4* and *L. plantarum* 6026 was higher than those of other combinations, which indicated that these two strains had a positive synergistic effect on enhancing the peptide yield. In line with our study, Cui et al. [13] reported that the peptide yield of soybean meal with mixed microorganisms was relatively higher compared with pure fermentation. In this study, the acid protease assisted the microorganisms in producing more soluble peptides than neutral protease, alkaline protease and flavor protease. Similarly, Jiang et al. [18] found that acid-protease-assisted LAB increased small peptide concentration.

The soluble peptide yield of the FRSMP was higher than that reported by Wang et al. [15] and Rong et al. [16], which was mainly related to the selections of microbial strains and the addition of exogenous protease. Different microorganisms produced different kinds of protease, while acid protease included exonucleases, and endonucleases, microbial protease and acid protease with multiple enzymatic hydrolysis sites increased the transformation from macro-molecular proteins into small molecular peptides, thereby synergistically cooperating to produce more small peptides and FAAs. Similarity, Bao, Zhang, Zheng, Ren and Lu [34] demonstrated that the peptide concentration of Spirulina platensis fermented with *B. subtilis* and *L. plantarum* was higher than those fermented with *B. subtilis* or *L. plantarum* only, and Cheng et al. [12] found that the soybean meal fermented with protease-assisting microorganisms had higher peptide concentration than that fermented by microorganisms only.

#### 4.1.2. Discussion on the Optimization of Fermentation Conditions

The underlying mechanisms affecting the small peptide yield depended on fermentation conditions, such as exogenous protease amount, moisture, inoculation size, cultivation medium composition, cultivation time and cultivation temperature. The variables (acid protease, fermentation time and moisture) were used for further optimization due to significant influences on rapeseed peptide yield obtained by PBD. According to the linear regression equation, the peptide yield increased with the addition of exogenous protease, which was due to the acid protease secreted from *Aspergillus niger* with a high capacity of hydrolyzing protein, and Jiang et al. [18] obtained the same conclusion. Different protease from different microorganisms had different cleavage sites of peptide bonds, therefore the additional increase in acid protease obtained from *Aspergillus niger* could facilitate *B. subtilis YY-4* and *L. plantarum 6026* to produce more peptides. Jiang et al. [18] reported that corn gluten meal fermented with acid protease significantly increased the small peptide content, reducing power and the DPPH, hydroxyl and superoxide anion radical scavenging capacity.

In addition, fermentation time was also a key factor to optimize and control the process. In this study, the small MW peptide yield of FRSMP increased with fermentation time according to the linear regression equation. In line with our result, Yang et al. [35] found that the small peptide content of the soybean meal fermented by *Bacillus amyloliquefaciens* increased with the increase in fermentation time, and Małgorzata et al. [36] also reported a greater improvement in the peptide content with 72 h fermentation (12.6%) in comparison with 48 h fermentation (6.8%) in lupin meal fermented by *Candida utilis.* Wang [37] reported that the soluble peptide content of fermented extrusion rapeseed meal reached the highest at 60 h, and it was consistent with the antioxidant activity in vivo and in vitro.

The small peptide yield of FRSMP was associated with moisture. Idris, Pandey, Rao and Sukumaran [38] reported that high moisture could affect the normal growth of bacteria by limiting the role of oxygen. Jiang et al. [31] demonstrated that corn gluten meal (MW < 10 kDa) fermented with BS5480 containing 46% moisture exhibited high peptide yield and antioxidant capacity in vivo.

In this study, the soluble peptide yield of FRSMP reached 230.56 mg/g, which was higher than the results reported by Xing et al. [39] and Wang et al. [15]. Several factors led to the higher peptide yield of FRSMP, as follows: the *B. subtilis YY-4* was screened by measuring the capacity of hydrolyzing rapeseed proteins; the combination of strains was screened by measuring the yield of rapeseed peptides; the combination of strains and protease was screened by measuring the yield of rapeseed peptides; and the fermentation conditions were optimized by the synthesis application of PBD and BBD, thereby producing a higher yield of rapeseed peptides. The essence of the novel fermentation was to produce more enzymes (various protease, cellulase, hemicellulase and glycosidase), and the complex structures including glycosidic bonds between some proteins and NDF were disrupted to increase the contact area, which enhanced the yields and efficiencies of microbial protease and acid protease.

Response surface curves were plotted to better influence the relationship between the variables and responses, and the optimal level in peptide yield was obtained within the selected range, which indicated that the selection range of the variable levels was reasonable.

### 4.2. Discussion on the Changes of the Nutritional Values of the RSM after Fermentation

The glucosinolate concentration of the FRSMP decreased to 6.15 μ mol/g, and it was safe as functional food/feed, which was lower than the result proposed by Wang et al. [15]. The drastic reduction in glucosinolate might be due to fermentation with acid-protease-assisting microorganisms, which produced more small peptides and FAAs for the growth and proliferation of microorganisms, thereby further hydrolyzing the glucosinolate. In this study, the NDF concentration of the FRSMP was decreased by 35.39%, and the result was consistent with the loose surface structure of the SEM image (Figure 4). Likewise, Li et al. [17] reported that the crude fiber content decreased in whole soybean flour (38.6%) during the fermentation with *L. casei*, as the bacteria could synthetize cellulose and hemicellulose. Moreover, the glucosinolate and NDF decomposed by microorganisms could be used as carbon sources to promote microbial proliferation, thereby improving the efficiency of microbial protease. Couto and Sanroman [40] reported that the efficiency of microbial protease can be improved by the removal of ANFs.

Phytic acid, as one of anti-nutritional factors, was formed by the esterification reaction of inositol and six phosphoric acids, and it had a high capacity of Fe^2+^ chelating, like EDTA. In this study, the phytic acid content of RSM after fermentation substantially decreased (Y = −0.1818x + 2.3154, R^2^ = 0.991), which meant that the availability of the phosphate increased due to the production of phytase during the novel fermentation. In addition, the increased amount of antioxidant activity in vitro (Fe^2+^ chelating activity) of the RSM after fermentation was strongly biased due to the high phytic acid content of the RSM, and the decrease in the phytic acid of the FRSMP indicated that the differences in Fe^2+^ chelating activity between the RSM and FRSMP were higher than the determined value, which indicated that the peptides and amino acids of the FRSMP had higher Fe^2+^ chelating activity than those of the RSM.

In this study, the CP concentration increase in the FRSMP was mainly due to producing gas, and the additions of exogenous protease and microorganisms enhanced the CP concentration to a certain degree. The decrease in fiber content was also an important reason for the increment of CP concentration, which was because the decomposed fiber could serve as an energy source for growth, and some of it had been bioconverted into protein [41]. In addition, the CP concentration of co-fermentation might increase, and Olukomaiya et al. [42] found that the protein content of lupin flour fermented by *Aspergillus sojae* and *Aspergillus ficuum* suffered a larger increase in comparison with individual strain fermentation.

In this study, total phenolic content of the FRSMP had no change compared to that of the RSM. Polanowska et al. [43] also reported a decreasing trend of total phenolic content during fermentation of faba beans; indeed, the increases in total phenolic content of the substrates after fermentation were common [44,45]. The increase or decrease in total phenolic content mainly depended on the microorganism strains, and Li et al. [46] detected a higher total phenolic content in kiwifruit wine by two *S. cerevisiae* strains compared to control, but the opposite result was found in four other strains. In addition to microorganisms, the type of phenolic content of the substrates was an important factor, and the combined action of the microorganisms and substrates resulted in different results in total phenolic content. During the fermentation, the increase in total phenolic content might be due to that the production of protease, esterase and glycosidase could release the bound phenolic compounds linked to the cell wall, thereby resulting in increased phydroxybenzoic acid and flavonol content; the decrease in total phenolic content might be due to the decompositions of some flavonol compounds of the substrate, and Bautista Exposito et al. [47] reported that the flavan-3ols of lentil flour with *L. plantarum* were decomposed.

### 4.3. Discussion on the Change of the Molecular Weight Distribution and Free Amino Acid Composition of the RSM after Fermentation

The molecular weight (length of the peptides) was an important parameter reflecting the solubility and bioactivity. The proteins of the RSM mainly included two main storage proteins (cruciferin, 12 s globulin; napin, 2 s albumin) and some of the smaller proteins, such as insulin inhibitors and lipoprotein. The average molecular weight of 12 s globulin was 300–310 kDa (six 50 kDa subunits, formed by heterodimer of 30 kDa and 20 kDa), and the average molecular weight of 2 s albumin was 12.5–14.5 kDa [48]. In this study, most proteins of the FRSMP were degraded to the soluble peptides with low molecular weight (less than 9.5 kDa) determined by SDS-PAGE, and the small peptides with low molecular weight (less than 1 k Da) analyzed by the ultrafiltration method significantly increased. In line with our study, Jiang et al. [14] found that fermentation with acid-protease-assisting *Lactobacillus* degraded the corn gluten–wheat bran mixture protein into low molecular weight peptides. The increased peptide concentration was consistent with high antioxidant ability, especially in small peptides, and many studies demonstrated that the small peptides with low molecular weight (less than 1 k Da) exhibited high bioactivity [49,50]. Adeola et al. [11] demonstrated that the small peptides with a low molecular weight (MW) (less than 3000 Da) display high levels of antioxidant activity. Wang et al. [15] reported that the small peptide content of the rapeseed meal fermented with mixed strains increased 4.54-fold, and Yang et al. [35] found that the soluble peptides (MW < 3 kDa) of the soybean meal byproduct after fermentation with *B. amyloliquefaciens* SWJS22 increased up to 40% from 14.66%.

The FAA was another important parameter reflecting the bioactivity in addition to the molecular weight. Li et al. [19] found whole soybean flour fermented by *Lactobacillus casei* markedly improved the FAA concentrations, and Liu et al. [46] reported that the total concentration of FAAs of fermented defatted wheat germ with *B. subtilis* increased as fermentation time increased. Bao, Zhang, Zheng, Ren and Lu [34] demonstrated that fermentation with *B. subtilis* and *L. plantarum* retained more total FAAs than *B. subtilis or L. plantarum*. The presence of different microorganisms modulated the yield of FAAs. In this study, the concentrations of the valine, leucine, phenylalanine and tryptophan increased during the process of fermentation. In line with our result, Li et al. [51] reported that these amino acids were positively correlated with the antioxidant activity.

Additionally, the concentration of free EAA of the FRSMP increased in the present study. Similarly, Yin et al. [52] reported a higher proportion of EAA in the FAAs of *L. plantarum NCU137* fermented adlay seed, and Bao, Zhang, Zheng, Ren and Lu [35] found that the fermentation with *B. subtilis* and *L. plantarum* increased 1.5-fold in the proportion of EAAs in the FAAs and accumulated more EAAs than the fermentation with *B. subtilis* or *L. plantarum.* In addition, the concentrations of the Glu and Lys markedly increased, which was beneficial to increasing the freshness and palatability.

### 4.4. Discussion on the Availability of the RSM after Fermentation

In the present study, the ADG values of weaned rats fed FRSMP were increased, which was consistent with previous studies where the growth performances of weaned piglets were enhanced by adding FSBM [53] and those of weaned calves were improved by adding fermented corn gluten–wheat bran with acid-protease-assisting *Lactobacillus* [17]. The growth rate is largely regulated by the GH-GF-1 axis, and GH is regulated by thyroxine (especially T_3_). In this study, the serum T_3_ and IGF-1 levels of weaned rats fed FRSMP increased compared to those fed RSM, and the tissue IGF-1 levels were elevated, which indicated that the FRSMP had high availability. The high availability of small peptides was due to high absorption in intestine: one main reason was the high efficiency of peptide carriers compared to amino acid carriers, and another important reason was that short-chained peptides can easily migrate into the interface between oil and water [48], which was also reflected by the increase in CP digestibility in vitro of the RSM after fermentation. In addition, the improvement of the palatability during the fermentation (the decrease in anti-nutritional factors, the increases in flavor amino acids and lactic acid) increased the intake of the diets, thereby enhancing the CP deposition rate to promote body growth.

### 4.5. Discussion on the Antioxidant In Vitro of the RSM after Fermentation

In this study, the DPPH scavenging activity, Fe^2+^ chelating activity and reducing power were used to evaluate the antioxidant activity of rapeseed peptides. Generally, the methods for evaluating antioxidant activity can be divided into two major groups according to their reaction mechanisms: hydrogen atom transfer based method (HAT) and single electron transfer based method (SET). The antioxidant mode of action of the DPPH method was SET, which can transfer one electron to reduce any compound [54]. The antioxidant mode of the reducing power includes SET and HAT, and it can react with the free radicals to terminate the chain reaction, and it is closely related to the hydrogen-donating ability of the contained reductones (HAT) [55]. Fe^2+^ can trigger free radical formation that causes lipid peroxidation, and the polypeptides can form a stable ring to chelate Fe^2+^, which serve as hydrogen donors to maintain the original valence (HAT) [56].

In this study, the DPPH scavenging activity, Fe^2+^ chelating activity and reducing power of the FRSMP were improved compared to those of the RSM, which indicated that the antioxidant mode of action of the rapeseed peptides included HAT and SET. The scavenge capacity of free radicals mainly depended on the type, chain length, composition sequence and position of amino acids. Antioxidant peptides containing Tyr residues mainly played their antioxidant role through the mechanism of HAT, and the antioxidant peptides containing Trp, Cys and His mainly deactivated free radicals by the mechanism of SET [57].

In this study, the RSM after fermentation had no significant change in total phenolic content, which indicated that the increase in antioxidant activity of the FERSMP mainly depended on the yield of rapeseed peptides with antioxidant activity. The antioxidant activity of FRSMP was increased. In line with our result, Liu [46] demonstrated that the increase with fermentation time of small peptides of defatted wheat germ fermented with *B. subtilis* was strongly related to antioxidant activity (R^2^ = 0.97). Ruann and Hélia [58] reported that the functional properties of the whey protein hydrolyzed by acid protease markedly increased, and Jiang et al. [17] found that the increased antioxidant capacity might be attributed to the fermentation with acid-protease-assisting *Lactobacillus*.

In this study, the DPPH scavenging activities of the FRSMP were linearly related to concentrations (R^2^ = 0.99) within the scope of concentration (0–1 mg/mL), and the DPPH scavenging activities of the FRSMP at different concentrations were higher than those of the RSM, which was consistent with the result of Li et al. [16]. The EC_50_ value of the DPPH scavenging activity was adopted to compare with various studies [10], and the EC_50_ value (480 µg/mL) of the rapeseed peptides from FRSMP was lower than that (710 µg/mL) reported by Pan, Jiang and Pan [59]. This discrepancy might be due to the rapeseed peptides from the RSM fermented with acid-protease-assisting *B. subtilis* YY-4 and *L. plantarum* 6026. The increased essence of the DPPH scavenging activities of the FRSMP might be due to producing more small molecular peptides and FAAs, which donated electrons to convert the DPPH radicals into harmless products, preventing the accumulation of the free radicals. The small molecular peptides were reported to exhibit better radical scavenging activities than the high molecular peptides. Rong et al. [16] reported that the DPPH radical scavenging activity of the rapeseed peptides with low molecular weights (610 Da) reached 37.19% at 52.64 µg/mL and that of the rapeseed peptides with high molecular weights (5500 Da) reached 17.21% at 94.37 ug/mL. Furthermore, the concentrations of the FAAs (tyrosine, cysteine and methionine) with radial scavenging activities of the FRSMP were more than those of the RSM, which may be another reason for the increase in DPPH scavenging activity.

Ferrous ions can promote the generation of active oxygen (hydroxyl radicals and peroxide radicals) to accelerate the lipid peroxide, and the peptides might be used as indirect antioxidant agents to inhibit the lipid peroxide by chelating ferrous ions. In this study, the RSM at a concentration of 6 mg/mL had the same ferrous-ion-chelating ability as FRSMP at a concentration of 3.25 mg/mL, which indicated that the FRSMP had a higher ferrous-ion-chelating activity compared to the RSM. The EC_50_ value of the ferrous-ion-chelating activity of the FRSMP was 2.75 mg/mL, which was lower than that of fermented rapeseed meal (4.9 mg/mL) [16] and that of fermented black soybean (207 mg/mL) [60], which indicated that the FRSMP had a stronger ferrous-ion-chelating activity.

In this study, the rapeseed peptide yield of FRSMP was 0.64 mg/mL when the absorbance was 1, and the value was lower than the result obtained by Rong et al. [16], which indicated that the reducing power of the peptides from the FRSMP was high. The increase in the reducing powers might be because the concentrations of certain amino acids with reducing power (methionine and glutamate) of the FRSMP increased compared to those of the RSM.

### 4.6. Discussion on the Immunomodulation Activity In Vitro of the RSM after Fermentation

Weaning is one of the most critical periods in an animal’s life, during which they have to cope with psychological, environmental and nutritional stressors, and these can lead to further oxidative stress which induces inflammation [61]. In this study, the increase in serum IgM and IgG concentration of weaned rats fed FRSMP indicated that the immune function was improved, and the increase in tissue SigA suggested that the intestinal barrier function was improved by weakening the surface hydrophobicity of bacteria and toxins. The results were consistent with many studies: Xu et al. [62] reported that the serum concentrations of IgM and IgG of the broiler chicken fed with fermented feed were increased, and Tang et al. [63] found that the fermented cotton meal increased serum IgG and IgM levels compared with the control diet. The immune function improvement of weaned rats fed FRSMP might correlate with the increase in the CP utilization, which was consistent with the high absorption of small peptides and FAAs. In line with our result, Ma, Shang, Wang, Hu and Piao [64] demonstrated that the weaning stress was alleviated by the increase in the absorbable protein as a synthetic material of the immunoglobulin, and Cheng et al. [11] also found that the immune functions of broilers fed the FSBM with more small peptides and FAAs were improved.

The pro-inflammatory cytokine (IL-6) levels of the serum and jejunum tissues of weaned rats fed FRSMP decreased, which indicated that the FRSMP had the potential to decrease the inflammatory response to increase the immune capacity. Wang et al. [55] demonstrated that inflammatory injury was relieved in weaned piglets fed FRSM by down-regulating the expression of pro-inflammatory cytokines, and Luti et al. [65] also found that sourdough fermented with selected LAB exhibited high antioxidant activity and anti-inflammatory activity.

## 5. Conclusions

In conclusion, the RSM fermented with acid-protease-assisting *L. plantarum* 6026 and *B. subtilis* YY-4 could degrade the rapeseed protein to small molecular peptides, which was demonstrated by the reduction in the intensity of the protein band on SDS-PAGE, the decrease in the peak area on the gel filtrate and the decline of the small peptide fraction in ultrafiltration. The antioxidant activities in vivo and in vitro of the RSM after fermentation with acid-protease-assisting mixed stains were increased, and the availability in vivo and in vitro was improved. Moreover, the immunomodulatory activity of the RSM after fermentation with acid-protease-assisting *L. plantarum* 6026 and *B. subtilis* YY-4 was enhanced. The removal of these anti-nutritional factors makes the rapeseed protein available in food applications. Overall, the study provided an innovative method to develop a functional food with high peptide content, which provided a feasible solution for the application of rapeseed meal in food, thereby clearly pointing to the direction of producing more active peptides and greatly enhancing the added value.

In the future, mapping and characteristics of the rapeseed peptides will be analyzed by the peptidome. The relationship at the molecular level between the structure properties and antioxidant activity in rapeseed peptides will be well established based on the aid of computer programs, and the peptides with high antioxidant activity will be separated and purified. Although the functionality of the FRSMP has been evaluated by in vitro and in vivo experiments, the application is still in the stage of laboratory research, which presents a certain gap from the commercial products on the market. To develop commercial products in food industry, the safety, gastrointestinal stability and potential sensory problems will be further investigated in a human clinical trial. On top of that, the fermentation products with high activity will be prepared into nanoparticle or microcapsules to apply as food additives and functional ingredients.

## Figures and Tables

**Figure 1 foods-11-02118-f001:**
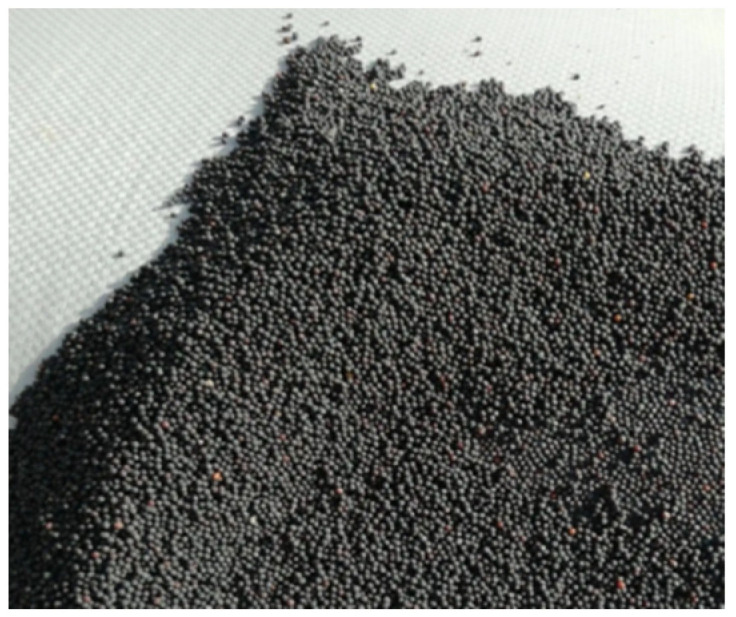
Rapeseed (Brassica). It is the seed of the rapeseed.of the cruciferous crop. The rapeseed is spherical or ellipsoid and is mainly used for oil extraction.

**Figure 2 foods-11-02118-f002:**
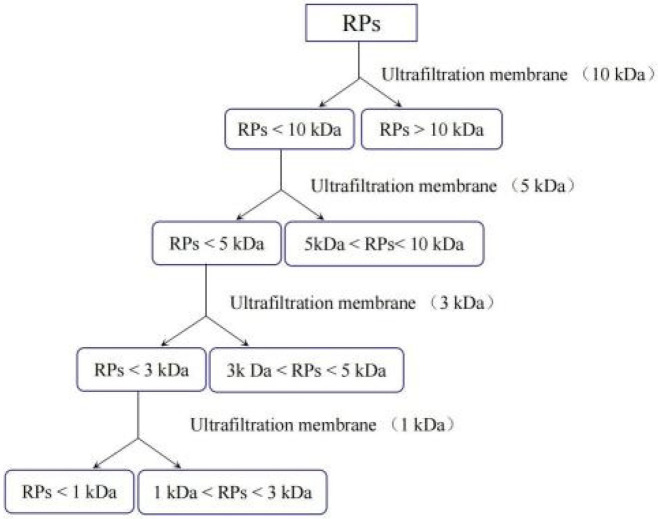
The ultrafiltration (UF) procedure. The permeate (<1 kDa) from the 1 kDa membrane (UFP-1-C-4M) was collected and lyophilized, while the retentate passed though the 3 kDa membrane (UFP-3-C-4M); the permeate (1–3 kDa) was collected to lyophilize and the retentate passed through the 5 kDa membrane (UFP-5-C-4M); the permeate (3–5 kDa) was collected to lyophilize and the retentate passed through the 10 kDa membrane (UFP-10-C-4M); the permeate (5–10 kDa) was collected to lyophilize, while the retentate (>10 kDa) was collected to lyophilize.

**Figure 3 foods-11-02118-f003:**
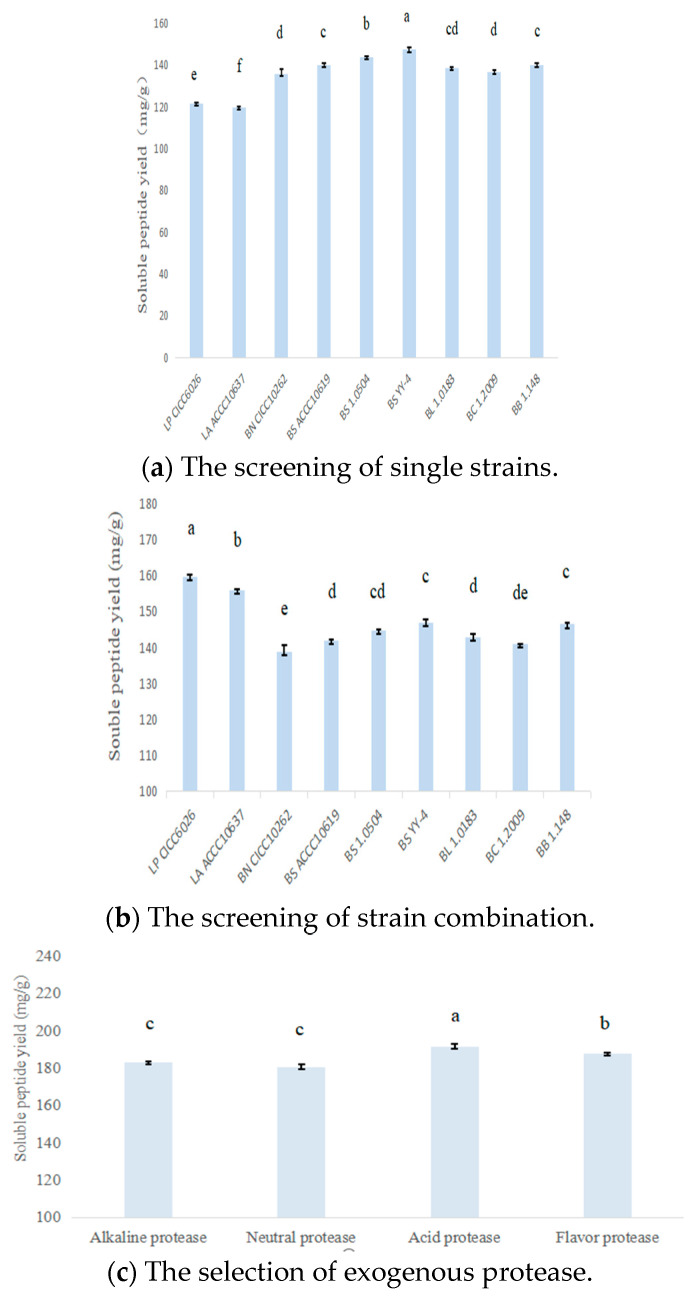
Screening of strain, strain combination and exogenous protease. Note: a–f, different lowercase superscripts indicate significant differences (*p* < 0.05) in the same row.

**Figure 4 foods-11-02118-f004:**
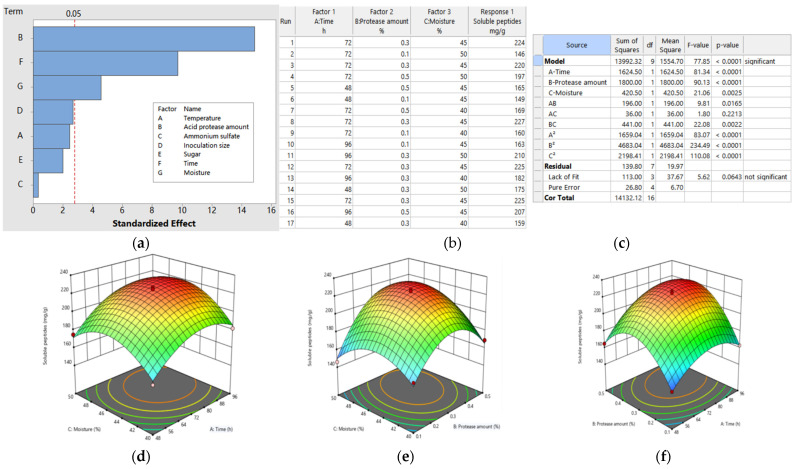
Optimization of fermentation conditions and response surface plots showing the interaction effect of variables on soluble peptide yield. (**a**) Pareto diagram describing the impact of factors (temperature, acid protease amount, ammonium sulfate, inoculation size, sugar, time and moisture) evaluated by a Plackett–Burman design on soluble peptide yield (mg/g); (**b**) Box Behnken design matrix for optimization of soluble peptide yield (mg/g); (**c**) analysis of variance (ANOVA) for response surface quadratic model on soluble peptide yield (mg/g); (**d**) interaction of moisture (%) and time©); (**e**) interaction of moisture (%) and acid protease amount (%); (**f**) interaction of acid protease amount (%) and time (h).

**Figure 5 foods-11-02118-f005:**
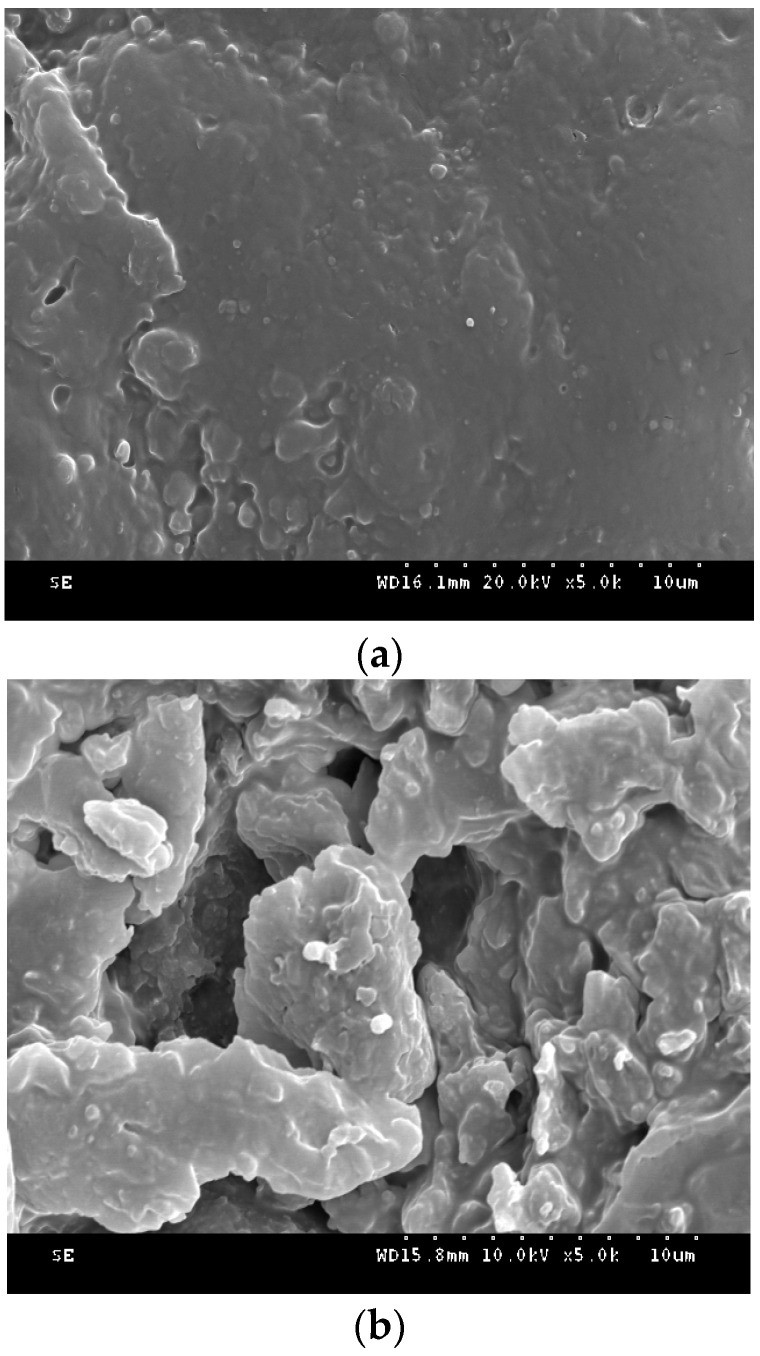
Scanning electron microscope images of the rapeseed meal (RSM) (**a**) and fermented rapeseed meal with acid protease (FRSMP) (**b**) (10 µm) at 5000× magnification. The samples were photographed with secondary electron image at a low acceleration voltage of 5.0 kV.

**Figure 6 foods-11-02118-f006:**
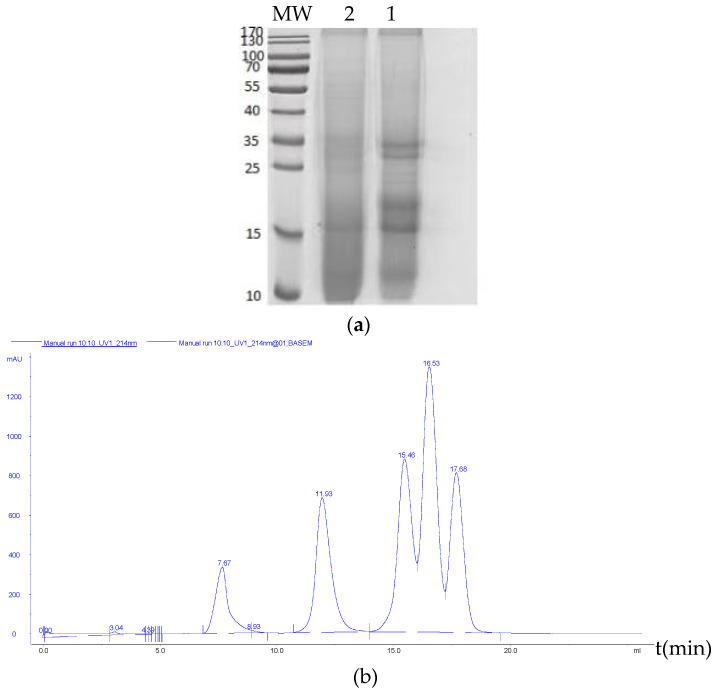
Weight distribution of rapeseed meal (RSM) and fermented rapeseed meal with acid protease (FRSMP): (**a**) SDS-PAGE pattern of the RSM and FRSMP (1, RSM; 2, FRSMP); (**b**) standard curve, the ordinate is Height (mAu); (**c**) weight distribution of RSM, the ordinate is Height (mAu); (**d**) weight distribution of FRSMP, the ordinate is Height (mAu).

**Figure 7 foods-11-02118-f007:**
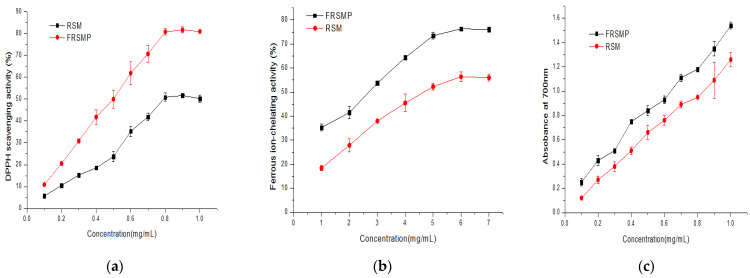
Antioxidant activities of the rapeseed meal (RSM) and fermented rapeseed meal with acid protease (FRSMP). (**a**) DPPH scavenging activity; (**b**) ferrous-ion-chelating activity; (**c**) reducing power.

**Table 1 foods-11-02118-t001:** Ingredients and compositions and nutrient levels of the control diet and experimental diet.

Item	Control Diet	Experiment Diet
Compositions (g/kg, as fed basis) Soybean meal	180	180
RSM	50	−
FERSM	−	50
Corn	390	390
Wheat	310	310
Fish meal	30	30
Stone	15	15
Salt	5	5
Oil	10	10
Premix ^1^	10	10
Nutrient level (%)		
CP	19.22	19.359
Energy	3.796	3.796
Fiber	2.88	2.565
Fat	3.49	3.495
Ca	0.274	0.275
*p*	0.482	0.486
Methionine	0.322	0.322
Lysine	0.891	0.896
Threonine Trp	0.694 0.219	0.694 0.22

^1^ Note: The premix provided the following per kilogram of diet: vitamin A, 14,000 IU; vitamin D3, 1500 IU; vitamin E, 5 mg; vitamin K3, 5 mg; vitamin B1, 13 mg; vitamin B2, 12 mg; vitamin B6, 12 mg; vitamin B12, 22 μg; pantothenic acid, 24 mg; nicotinic acid, 60 mg; folic acid, 6 mg; biotin, 200 μg; choline chloride, 350 mg; Cu, 10 mg; Mn, 75 mg; Zn, 30 mg; Fe, 120 mg; I, 0.5 mg; Se, 0.2 mg.

**Table 2 foods-11-02118-t002:** Chemical composition and peptide fractions of the RSM and FRSMP.

Composition	RSM	FRSMP	SEM	*p*
Glucosinolate (µmol/g DM)	75.17 ^a^	6.15 ^b^	15.44	0.000
Crude protein(mg/g DM)	360 ^b^	405 ^a^	11.3	0.017
NDF (mg/g DM)	226 ^a^	140 ^b^	19.5	0.000
Lactic acid (g/kg DM)	30.50 ^b^	110.30 ^a^	24.43	0.000
Soluble peptides (mg/g DM)	53.33 ^b^	231.05 ^a^	30.85	0.000
Total phenolic content (µg/g DM)	2.18	2.45	0.039	0.579
Phytic acid (µg/gDM)	2.68 ^a^	0.25 ^b^	0.544	0.000
Peptide fraction (kDa) (mg/g DM)				
<1 kDa	5.39 ^b^	80.10 ^a^	16.70	0.000
1–3 kDa	0.44 ^b^	2.14 ^a^	0.38	0.000
3–5 kDa	2.85 ^b^	32.99 ^a^	6.74	0.000
5–10 kDa	2.59 ^b^	56.51 ^a^	12.06	0.000
>10 kDa	42.08 ^a^	19.74 ^b^	5.00	0.000

Note: RSM, rapeseed meal; FRSMP, fermented rapeseed meal with microorganism and protease; different lowercase superscripts indicate significant differences (*p* < 0.05) in the same row; SEM, standard error; *p*-value of significance.

**Table 3 foods-11-02118-t003:** Compositions and concentrations of free amino acids of the RSM and FRSMP (ng/g DM).

Composition	RSM	FRSMP	SEM	*p*
Glycine	26.03 ^b^	79.02 ^a^	11.87	0.000
Alanine	396.95 ^a^	318.18 ^b^	17.65	0.000
Serine	98.31 ^a^	28.05 ^b^	15.73	0.000
Proline	529.27 ^b^	673.09 ^a^	32.29	0.000
Valine	182.34 ^a^	389.77 ^b^	46.42	0.000
Threonine	107.84 ^a^	90.69 ^b^	4.21	0.012
Cysteine	6.33 ^b^	17.83 ^a^	2.61	0.000
Isoleucine	79.34 ^a^	281.1 ^b^	45.14	0.000
Asparagine	219.76 ^a^	31.04 ^b^	42.21	0.000
Aspartic acid	402.52 ^a^	167.63 ^b^	52.61	0.000
Glutamine	37.13 ^a^	26.46 ^b^	2.52	0.004
Glutamate	708.95 ^b^	2052.08 ^a^	300.57	0.000
Methionine	3.73 ^b^	454.33 ^a^	100.79	0.000
Histidine	21.99	24.64	0.78	0.083
Phenylalanine	99.25 ^b^	627.23 ^a^	118.15	0.000
Arginine	318.19 ^a^	111.00 ^b^	46.35	0.000
Tryptophan	85.64 ^b^	171.22 ^a^	19.16	0.000
Lysine	154.15 ^b^	1261.30 ^a^	247.71	0.000
Tyrosine	50.05 ^b^	500.66 ^a^	100.78	0.000
Leucine	103.02 ^b^	608.99 ^a^	113.14	0.000
Total amino acids	3630.80 ^a^	7914.31 ^b^	958.35	0.000
EAA	815.32 ^b^	3884.63 ^a^	686.53	0.000
EAA/NEAA	0.29 ^b^	0.97 ^a^	0.059	0.000

Note: RSM, rapeseed meal; FRSMP, fermented rapeseed meal with microorganism and protease; different lowercase superscripts indicate significant differences in the same row (*p* < 0.05); SEM, standard error; *p*-value of significance; EAA, essential amino acid; NEAA, non-essential amino acid.

**Table 4 foods-11-02118-t004:** Effects of the RSM before and after fermentation on nutrient digestibility in vitro and growth-promoting activities of weaned rats in vivo.

Item	RSM	FRSMP	SEM	*p*
DMD (g/kg)	700 ^b^	779 ^a^	17.92	0.000
CPD (g/kg)	861 ^b^	880 ^a^	4.67	0.012
DMI (g/d)	18.06 ^a^	18.76 ^a^	0.28	0.088
ADG (g/d)	7.56 ^b^	8.46 ^a^	0.24	0.034
FCR	2.39	2.22	0.05	0.115
**Serum**				
GH (ng/mL)	5.19	6.15	0.24	0.260
IGF-1 (ng/mL)	167.11 ^a^	204.75 ^b^	5.86	0.011
T_3_ (ng/mL)	0.46 ^b^	0.60 ^a^	0.03	0.014
T_4_ (ng/mL)	45.39 ^b^	66.16 ^a^	4.49	0.004
**Tissue**				
IGF-1 (ng/mg)	10.82 ^b^	17.02 ^a^	0.89	0.000

Note: RSM, rapeseed meal; FRSMP, fermented rapeseed meal with microorganism and protease; different lowercase superscripts indicate significant differences in the same row (*p* < 0.05); SEM, standard error; *p*-value of significance.

**Table 5 foods-11-02118-t005:** Effects of the RSM and FPRSM on immunomodulation activity of the weaned rats in vivo.

Item	RSM	FRSMP	SEM	*p*
Serum				
IgA (g/L)	0.49	0.56	0.02	0.087
IgM (g/L)	0.42 ^b^	0.55 ^a^	0.02	0.000
IgG (g/L)	5.45 ^b^	6.33 ^a^	0.16	0.028
IL-1β (pg/mL)	38.67 ^a^	34.32 ^b^	1.32	0.089
IL-6 (pg/mL)	227.93 ^a^	155.68 ^b^	8.50	0.000
Tissue				
SigA (µg/mg)	0.54 ^b^	0.68 ^a^	0.02	0.039
IL-1β (pg/mg)	3.32	2.96	0.09	0.316
IL-6 (pg/mg)	22.65 ^a^	16.99 ^b^	0.68	0.000

Note: RSM, rapeseed meal; FRSMP, fermented rapeseed meal with protease; different lowercase superscripts indicate significant differences in the same row (*p* < 0.05); SEM, standard error; *p*-value of significance.

## Data Availability

None of the data were deposited in an official repository. Data is contained within the article.

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
