# Peer review of "A Novel Fermented Rapeseed Meal, Inoculated with Selected Protease-Assisting Screened B. subtilis YY-4 and L. plantarum 6026, Showed High Availability and Strong Antioxidant and Immunomodulation Potential Capacity"

_foods, 2022, doi:10.3390/foods11142118_

Round 1

Reviewer 1 Report

Comments

1. 11-14, first sentence in abstract need to be modified and divided in to two for more clarity to readers.

2. 15-17, whether small peptides were different from soluble peptides, if not, both sentences can be merged.

3. 19, what was the level of bioactivity – mild, moderate or high.

4. 22, FRSMP can be added as a keyword.

5. 26, rapeseed figure can be included in the introduction part.

6. 54, 58, 73, 548 “et al.,” OR “et al.” – it should be uniform as “et al.” in whole manuscript.

7. 59, 70, 75, Bacillus subtilis can be written as B. subtilis, as we already used abbreviated form in the text.

8. 177, what the symbol represents…. PE/7.5/300 (?7.5×300 mm)

9. 181, 529, terms like v/v/v & in vitro should be in italics.

10. 209, 254, 267, 275, 290, 310, 318, 336, 343, 344, 364, 385, 386, 438, Tab.1, Tab.2, Tab.3, Tab.4  and Tab.5 can be mentioned as Table 1, table 2, Table 3, Table 4 and Table 5.

11. 271, gaps between words can be adjusted further.

12. 316, (10 ?m) need to be rechecked.

13. 327, Table 2, Total phenolic content (?g/g DM); Phytic acid (?g/g DM); Glucosinolate (?mol/g DM) need to be rechecked.

14. 351, immune activity should be written as immunomodulation activity.

15. 366 & 378, 7914.31 ?g/g DM & FRSMP (?g/g DM) need to be rechecked.

16. 407, 408, 633, ?g/ml need to be rechecked.

17. 443, Table 5, SigA (?g/mg) need to be rechecked.

18. 480, “Jiang, et al.,” should be written as “Jiang et al.”.

19. 558, flaven-3ols should be corrected.

20. 641, 643, mL OR ml, it should be uniform throughout the manuscript.

21. 709, 710, journal name, year, vol, issue and page numbers need to be rechecked.

22. 713, 822, abbreviation of journal need to be rechecked.

23. 730, 731, article number is missing.

24. 744, 782, 816, 882, ‘&’ is missing before last author.

25. 754, 806, 843, 862, journal appreviation need to be rechecked/corrected.

26. 757, volume number is missing.

27. 762, 771, 790, gap should be adjusted.

28. 764, journal name should be in italics.

29. 791, plantarum should be in italics.

30. 799, page numbers should be written correctly; issue number and doi number are missing.

31. 810, 814, there is no need to mention ‘16’ & ‘15’, these are dates of publication.

32.  814, year should be in bold.

33. 873, doi number is missing, page numbers need to be corrected, issue number can be included.

34. 876, doi and issue number can be included.

35. 883, year should be in bold.

36. 676, conclusion can be improved.

Author Response

Dear reviewer:

Thank you very much! We are truly grateful to concrete comments and helpful suggestions for revision, and we have revised the manuscript carefully according to your suggestions (point-by-point). Our point-by-point responses to the comments are as follow, and revised parts have been marked in yellow in the revised manuscript. The English has been improved by the native English speaker. If there is any question, you feel free to contact us, and we are very willing to improve our manuscript at all times until you are satisfied. Best regards for you. Everything goes well.

Sincerely yours,

Yan Wang

Address: College of Food and Biological Engineering, Qiqihar University, Qiqihar, 161006, P. R.

  1. 11-14, first sentence in abstract need to be modified and divided in to two for more clarity to readers.

Response: Thank you for your comments. According to your comment, the first sentence in abstract has been modified, and has been divided into two sentence in Line 11-15 of revised manuscript according to your comment. The changes have been highlighted in yellow.

  1. 15-17, whether small peptides were different from soluble peptides, if not, both sentences can be merged.

Response: Thank you for your comments. The small peptides refer to the small-molecular fraction of the soluble protein, generally small peptides refer the peptides with 3-10 amino acids. To avoid the misunderstanding, we have changed the writings “small peptides” to “small-molecular peptides” in Line 17 of revised manuscript. The change have been highlighted in yellow.

  1. 19, what was the level of bioactivity – mild, moderate or high.

Response: Thank you for your comments. According to your comment, the levels of antioxidant activity and immunomodulation activity should be high, so we have added the level in Line 19 of revised manuscript. The changes have been highlighted in yellow.

  1. 22, FRSMP can be added as a keyword.

Response: Thank you for your comments. According to your comment, the word “FRSMP” has been added as a key word in Line 22 of revised manuscript. The change have been highlighted in yellow.

  1. 26, rapeseed figure can be included in the introduction part.

Response: Thank you for your comments. According to your comment, the rapeseed figure (Figure 1) has been included in the introduction part, and has been added in Line 44-45 of revised manuscript.  

  1. 54, 58, 73, 548 “et al.,” OR “et al.” – it should be uniform as “et al.” in whole manuscript.

Response: Thank you for your comments. The writing “et al.,” in Line 54, 58, 73, 548 of original manuscript have been changed to “et al.” in Line 64, 68, 84 and 572 according to your comment. According to your comment, we checked the whole manuscript, and all the writings (et al.,) have been changed to the writings (et al.) to keep uniform.

  1. 59, 70, 75, Bacillus subtiliscan be written as subtilis, as we already used abbreviated form in the text.

Response: Thank you for your comments. According to your comment, the writings (Bacillus subtilis) in Line 59, 70, 75 of original manuscript have been written as B. subtilis in Line 70, 80, 86-87 of revised manuscript. The writing (Bacillus subtilis) has been checked in whole manuscript carefully, and the writing “Bacillus subtilis has been changed to “B. subtilis” in Line 69 of revised manuscript. All the changes have been highlighted in yellow.

  1. 177, what the symbol represents…. PE/7.5/300 (?7.5×300 mm)

Response: Thank you for your comments. The symbol (Φ) stands for the inter diameter, and the abbreviate of the inter diameter is ID. Therefore the ID has been added in the Line 189 of revised manuscript. The change has been highlighted in yellow.

  1. 181, 529, terms like v/v/v & in vitro should be in italics.

Response: Thank you for your comment. The terms v/v/v & in vitro in Line 181 and 529 of original manuscript have been changed to italics “v/v/v” and “in vitro” in Line 194 and 199-200 of revised manuscript. And we have checked all the writings “in vitro” and “in vivo”, and changed to “in vivo” in Line 218 of revised manuscript. The changes have been highlighted in yellow.

  1. 209, 254, 267, 275, 290, 310, 318, 336, 343, 344, 364, 385, 386, 438, Tab.1, Tab.2, Tab.3, Tab.4 and Tab.5 can be mentioned as Table 1, table 2, Table 3, Table 4 and Table 5.

Response: Thank you for your comment. According to your comments, 209, 254, 267, 275, 290, 310, 318, 336, 343, 344, 364, 385, 386, 438, Tab.1, Tab.2, Tab.3, Tab.4 and Tab.5 have been changed to Table 1, Table 2, Table 3, Table 4 and Table 5, and all the changes have been highlighted in yellow. According to your comment (5), the Figure 1 (figure of rapeseed) has been added. And the Fig1, Fig2 (a, b), Fig 2(c), Fig 3(a), Fig3 (b, c), Fig3 (d, e, f), Fig 4, Fig 5 (a), Fig 5(b), Fig 5(c, d), Fig 6 (a), Fig 6(b), Fig 6(c) of the original manuscript have been changed to Figure 2, Figure 3(a, b), Figure 3(c), Figure 4(a), Figure4 (b, c), Figure4 (d, e, f), Figure 5, Figure 6 (a), Figure 6(b), Figure 6(c, d), Figure 7 (a), Figure 7(b), Figure 7(c) in Line 163, 264, 268, 282, 290, 306, 325, 352, 360, 361, 427, 437-438, 448 of the revised manuscript respectively. All the changes have been highlighted in yellow.  

  1. 271, gaps between words can be adjusted further.

Response: Thank you for your comment. According to your comment, the gaps between words have been adjusted further.

  1. 316, (10 ?m) need to be rechecked.

Response: Thank you for your comment, and it is important for the quality of the manuscript. The unit should been changed to “µ”, not “u”, and we have change the unit in Line 332 of revised manuscript. The change has been highlighted in yellow.

  1. 327, Table 2, Total phenolic content (?g/g DM); Phytic acid (?g/g DM); Glucosinolate (?mol/g DM) need to be rechecked.

Response: Thank you for your comment. According to your comment, the units of total phenolic content (ug/g DM), Phytic acid (ug/g DM) and Glucosinolate (umol/g DM) in Table 2 have been changed to total phenolic content (µg/g DM), Phytic acid (µg/g DM) and Glucosinolate (µmol/g DM). The changes have been highlighted in yellow.

  1. 351, immune activity should be written as immunomodulation activity.

Response: Thank you for your comment. According to your comment, the immune activity has been changed to the immunomodulation activity in Line 368 of revised manuscript.

  1. 366 & 378, 7914.31 ?g/g DM & FRSMP (?g/g DM) need to be rechecked.

Response: Thank you for your comment. We are sorry to make a mistake, and the units of 7914.31 ?g/g DM & FRSMP (?g/g DM) have been changed to ng/g DM in Line 389 and 401 of revised manuscript. The changes have been highlighted in yellow.

  1. 407, 408, 633, ?g/ml need to be rechecked.

Response: Thank you for your comment. According to your comment, the unit (ug/ml) in Line 407, 408 and 633 of original manuscript have been changed to (µg/ml) in Line 430, 431 and 682 of revised manuscript. The changes have been highlighted in yellow.

  1. 443, Table 5, SigA (?g/mg) need to be rechecked.

Response: Thank you for your comment. According to your comment, the unit of  SigA has been rechecked, and the unit “ug/mg” has been changed to “µg/mg” in Table 5, and the change has been highlighted in yellow .

  1. 480, “Jiang, et al.,” should be written as “Jiang et al.”.

Response: Thank you for your comment. According to your comment, the writing “Jiang, et al.,” in Line 480 of original manuscript has be written as “Jiang et al.” in Line 502 of revised manuscript, and the change has been highlighted in yellow.

  1. 558, flaven-3ols should be corrected.

Response: Thank you for your comment, and we are sorry to make a mistake. According to your comment, the writing “flaven-3ols” has been changed to flavan-3-ols in Line 589 of revised manuscript, and the change has been highlighted in yellow.

  1. 641, 643, mL OR ml, it should be uniform throughout the manuscript.

Response: Thank you for your comment. According to your comment, the unit (mL OR ml) has been checked throughout the manuscript. The units have been changed to the uniform unit (mL) throughout the manuscript.

  1. 709, 710, journal name, year, vol, issue and page numbers need to be rechecked.

Response: Thank you for your comment. We are sorry to make a mistake. The journal name, year, vol, issue and page numbers have been rechecked, and the year, vol, issue have been changed in Line 805 of revised manuscript [reference, 9], and highlighted in yellow.

  1. 713, 822, abbreviation of journal need to be rechecked.

Response: Thank you for your comment. According to your comment, the abbreviation of journal in Line 713 and 822 of original manuscript have been rechecked. The abbreviation of journal (Line 822 of original manuscript) has been changed to “Int. J. Mol. Sci.” in Line 947 of revised manuscript [reference, 51], and highlighted in yellow.

  1. 730, 731, article number is missing.

Response: Thank you for your comment. We are sorry to make a mistake, and the article number (108238) has been added in Line 826 of revised manuscript [reference, 15]. The change has been highlighted in yellow.

  1. 744, 782, 816, 882, ‘&’ is missing before last author.

Response: Thank you for your comment. We are sorry to make a mistake. According to your comment, the symbol ‘&’ has been added in Line 839, 877, 916 and 998 of revised manuscript [reference, 19, 31, 43, 66], and the changes have been highlighted in yellow.

  1. 754, 806, 843, 862, journal appreviation need to be rechecked/corrected.

Response: Thank you for your comment. According to your comment, the journal appreviations of Line 754, 806, 843, 862 of original manuscript have been checked.   The journal appreviations of Line 754, 806, 843, 862 of original manuscript have been changed to “Appl. Microbiol. Biot.”, “J. Tradit. Chin. Med.”, “J. Func. Foods”, and “J. Food Eng.” in Line 808 [reference, 10], 849 [reference, 22], 885 [reference, 33] and 910 [reference, 41] of revised manuscript. All the changes have been highlighted in yellow.

  1. 757, volume number is missing.

Response: Thank you for your comment. We are sorry to make a mistake. According to your comment, the volume number has been added in Line 852 [reference, 23] of revised manuscript. The change has been highlighted in yellow.

  1. 762, 771, 790, gap should be adjusted.

Response: Thank you for your comment. The gaps in 762, 771 and 790 of original manuscript have been adjusted according to your comment.

  1. 764, journal name should be in italics.

Response: Thank you for your comment. The journal name has been changed in italics in Line 860 of revised manuscript, and the change has been highlighted in yellow.

  1. 791, plantarum should be in italics.

Response: Thank you for your comment. The plantarum has been changed to italics in Line 885 of revised manuscript. The change has been highlighted in yellow.

  1. 799, page numbers should be written correctly; issue number and doi number are missing.

Response: Thank you for your comment. The page numbers have been corrected according your comment, and added in Line 893 [reference, 36] of revised manuscript. The issue number has been added in Line 893 of revised manuscript, and the doi number has been added in Line 894 of revised manuscript. The changes have been highlighted in yellow.

  1. 810, 814, there is no need to mention ‘16’ & ‘15’, these are dates of publication.

Response: Thank you for your comment. The “16” and “15” have been deleted in the  reference [42] of revised manuscript according to your comment.

  1. 814, year should be in bold.

Response: Thank you for your comment. The year has been bold in reference [42] (Line 914 of revised manuscript), and the change has been highlighted in yellow.

  1. 873, doi number is missing, page numbers need to be corrected, issue number can be included.

Response: Thank you for your comment. We are so sorry to make mistakes. According to your comment, the page numbers have been corrected in Line 988 [reference, 63] of revised manuscript, and issue number can be added in Line 987 of revised manuscript. The doi number has been added in Line 989 of revised manuscript. The changes have been highlighted in yellow.

  1. 876, doi and issue number can be included.

Response: Thank you for your comment. The doi number has been added in Line 993 [reference, 63] of revised manuscript. The “issue number” has been added in Line 992 of revised manuscript. The changes have been highlighted in yellow.

  1. 883, year should be in bold.

Response: Thank you for your comment. According to your comment, the year in the references has been changed to bold in Line 1000 [reference, 66] of revised manuscript, and highlighted in yellow.

  1. 676, conclusion can be improved.

Response: Thank you for your comment. The conclusion has been improved in Line 737-762 of revised manuscript according to your comment, and the changes have been highlighted in yellow.

I have updated my manuscripts. Please see the attachment.

Reviewer 2 Report

Dear Author, I reviewed the manuscript entitled A novel fermented rapeseed meal, inoculated with selected protease assisting screened B.substilis YY-4 and L.plantarum 6026, showed high availability, strong antioxidant and immunomodulation potential capacity. This manuscript presents relevant information about rapeseed meal bioactive properties. However, some sections of the presented data can be improved. For this reason, I considered that this manuscript needs minor changes . 

Additional comments.

Highlight the advantages of using rapeseed meal as functional food for its bioactive compounds content.

Check paragraphs extension in this manuscript.

Include an experimental design that contents statistical factors and variable of response in the statistical analyses applied to the findings of this research.

Include a possible antioxidant mode of action of rapeseed peptides in the tested protocols.

Try to compare the obtained findings with similar assays where microbial proteases were applied to enhance bioactive compounds content in rapessed or similar products. 

Include future trends to keep working with the obtained data. 

Try to conclude with a general statement of the most relevant part of this study.

Author Response

Dear reviewer:

  Thank you very much! We are truly grateful to concrete comments and helpful suggestions for revision, and we have revised the manuscript carefully according to your suggestions (point-by-point). Our point-by-point responses to the comments are as follow, and revised parts have been marked in yellow in the revised manuscript. The English has been improved by the native English speaker. If there is any question, you feel free to contact us, and we are very willing to improve our manuscript at all times until you are satisfied. Best regards for you. Everything goes well.

Sincerely yours,

Yan Wang

Address: College of Food and Biological Engineering, Qiqihar University, Qiqihar, 161006, P. R.

Dear Author, I reviewed the manuscript entitled A novel fermented rapeseed meal, inoculated with selected protease assisting screened B.substilis YY-4 and L.plantarum 6026, showed high availability, strong antioxidant and immunomodulation potential capacity. This manuscript presents relevant information about rapeseed meal bioactive properties. However, some sections of the presented data can be improved. For this reason, I considered that this manuscript needs minor changes . 

Additional comments.

Highlight the advantages of using rapeseed meal as functional food for its bioactive compounds content.

Response: Thank you for your comment. According to your comment, we have added the advantages of using rapeseed meal as functional food for its bioactive compounds in Line 28-30 (amino acid compositions) and Line 36-40 (antioxidant activity of rapeseed peptide) of revised manuscript. The changes have been highlighted in yellow. The advantages are as follows:

Rapeseed proteins have a well balanced amino acid profile with high quantities of indispensable amino acids, especially sulphur-containing amino acids (Citeau, Regis Carre Fine, 2016). Besides, the peptide (Pro-Ala-Gly-Pro-Phe) from rapeseed protein hydrolysate has high DPPH scavenging rate (Shao, Wang, Xu, & Gao, 2009). The rapeseed peptides hydrolyzed by flavourzyme and alcalase show a high reducing antioxidant power and hydroxyl radical scavenging ability, and exhibit a high inhibited activity with 50% in the serum MDA level (Xue, Yu, Liu, Wu, Kou, & Wang, 2009).

Check paragraphs extension in this manuscript.

Include an experimental design that contents statistical factors and variable of response in the statistical analyses applied to the findings of this research.

Response: Thank you for your comment. According to your comment, we have added the statistical factors and variable of response in the statistical analyses applied to the findings of this research. We have added the extension on addition of acid protease and soluble peptides applied to the findings of this research in Line 506-508 of revised manuscript, and added the extension on fermentation time and soluble peptides applied to the findings of this research in Line 516-518 of revised manuscript, and added the moisture and soluble peptides applied to the findings of this research in Line 521-523 of revised manuscript. The changes have been highlighted in yellow.

Include a possible antioxidant mode of action of rapeseed peptides in the tested protocols.

Response: Thank you for your comment. According to your comment, a possible antioxidant mode of action of rapeseed peptides has been discussed, and the discussion have been added in Line 648-667 of revised manuscript. The changes have highlighted in yellow.

In this study, the DPPH scavenging activity, Fe2+ chelating activity and reducing power were used to evaluate the antioxidant activity of rapeseed peptides. Generally, the methods for evaluating antioxidant activity can be divided into two major groups according to their reaction mechanisms: hydrogen atom transfer based method (HAT) and single electron transfer based method (SET). The antioxidant mode of action of the DPPH method was SET, which can transfer one electron to reduce any compound (Wen, Zhang, Zhang, Duan and Ma, 2020). The antioxidant mode of the reducing power includes SET and HAT, and it can react with the free radicals to terminate the chain reaction, and it is closely related to the hydrogen-donating ability of the contained reducones (HAT) (Yu et al., 2020). Fe2+ can trigger free radical formation and that caused lipid peroxidation, and the polypeptides can form a stable ring to chelate Fe2+, which served as hydrogen donors to maintain the original valence (HAT) (Himaya, Ryu, Ngo and Kim, 2012).

In this study, the DPPH scavenging activity, Fe2+ chelating activity and reducing power of the FRSMP were improved compared to those of the RSM, which indicated that the antioxidant mode of action of the rapeseed peptides included HAT and SET. The scavenge capacity of free radicals mainly depended on the type, chain length, composition sequence position of amino acids. Antioxidant peptides containing Tyr residues mainly played its antioxidant role through the methanism of HAT, and the antioxidant peptides containing Trp, Cys and His mainly deactivated free radicals by the mechanism of SET (Huang, Ou and Prior, 2005).  

Try to compare the obtained findings with similar assays where microbial proteases were applied to enhance bioactive compounds content in rapessed or similar products. 

Response: Thank you for your comment. According to your comment, “the obtained findings with similar assays where microbial proteases were applied to enhance bioactive compounds content in rapessed or similar products” has been added in the Line 607-612 of revised manuscript. Adeola et al. (2014) demonstrated that the small peptides with a low molecular weight (MW) (less than 3000Da) display high levels of antioxidant activity. Wang et al. (2019) reported that the small peptide content of the rapeseed meal fermented with mixed strains increased by 4.54 fold, and Yang et al. (2019) found that the soluble peptides content (MW<3000Da) of the soybean meal byproduct after fermentation with B. amyloliquefaciens SWJS22 increased up to 40% from 14.66%.

Include future trends to keep working with the obtained data. 

Response: Thank you for your comment. According to your comment, the future trends to keep working with the obtained data have been added in the conclusion (Line 751-762 of revised manuscript).  

In the future, the mapping and characteristics of the rapeseed peptides will be analyzed by the peptidome. The relationship at the molecular level between the structure properties and antioxidant activity in rapeseed peptides will be well established based on the aided computer, and the peptides with high antioxidant activity will be separated and purified. Although the functionality of the FRSMP has been evaluated by in vitro and in vivo experiment, however, the application is still in the stage of laboratory research, which presents a certain gap from the commercial products in market application. To develop the commercial products in food industry, the safety, gastrointestinal stability and potential sensory problems will be further  investigated by the human clinical trial. On top of that, the fermentation products with high activity will be prepared into nano particle or microcapsules to apply as food additives and functional ingredients.

Try to conclude with a general statement of the most relevant part of this study.

Response: Thank you for your comment. According to your comment, a general statement of the most relevant part has been added in Line 738-750 of revised manuscript.

The general statement was that the study provided an innovative method to develop a functional food ingredient with high peptide content, which provided a feasible solution for the application of rapeseed meal in the food, thereby pointing to the direction of producing more active peptides and greatly enhancing the value-added of rapeseed meal.

The updated manuscript has been added in the attachment. Please see the attachement. Thank you very much!

Reviewer 3 Report

Foods. Manuscript ID: foods-1778642

A novel fermented rapeseed meal, inoculated with selected protease assisting screened B.substilis YY-4 and L.plantarum 6026, showed high availability, strong antioxidant and immunomodulation potential capacity.

The paper well written and interesting focused the attention on the fermentation of rapeseed meal, a a novel fermentation to improve nutritional and biological properties.

In literature, many studies analyzed rapeseed meal and fermented rapeseed meal with Aspegillus as animal feed (pigs, mink), but this study focalized the attention on the combined used of two microrganisms to better produce small peptide with antioxidant, biactive function and above all with immunomodulation capacity.

So the study can add more informations about the use of rapaseed meal and give some innovative indications for the possible human use, even if, in my opinion, the use as functional food should be more investigated.

The paper is described in details, with chemical and biological analyses. The data are well discussed to confirm the initial proposal of a combined microrganisms to produce small peptide with good activity. The subdivision into paragraphs helps the reading and focuses attention on the subject of the paragraph.

If I can give a suggestion to the authors, in the conclusions I would better specify the possible use of the fermented product (how it can be used?? in which formulation it could be proposed???), to give more prominence to the possible human use, since many literature involves the use of fermented rapeseed for animal use.

The number of references is good and with recent date. Only the references n. 28-29-38- and 58 miss the year written in bold.

Author Response

Dear reviewer:

  Thank you very much! We are truly grateful to concrete comments and helpful suggestions for revision, and we have revised the manuscript carefully according to your suggestions (point-by-point). Our point-by-point responses to the comments are as follow, and revised parts have been marked in yellow in the revised manuscript. The English has been improved by the native English speaker. If there is any question, you feel free to contact us, and we are very willing to improve our manuscript at all times until you are satisfied. Best regards for you. Everything goes well.

Sincerely yours,

Yan Wang

Address: College of Food and Biological Engineering, Qiqihar University, Qiqihar, 161006, P. R.

Foods. Manuscript ID: foods-1778642

A novel fermented rapeseed meal, inoculated with selected protease assisting screened B.substilis YY-4 and L.plantarum 6026, showed high availability, strong antioxidant and immunomodulation potential capacity.

The paper well written and interesting focused the attention on the fermentation of rapeseed meal, a a novel fermentation to improve nutritional and biological properties.

In literature, many studies analyzed rapeseed meal and fermented rapeseed meal with Aspegillus as animal feed (pigs, mink), but this study focalized the attention on the combined used of two microrganisms to better produce small peptide with antioxidant, biactive function and above all with immunomodulation capacity.

So the study can add more informations about the use of rapaseed meal and give some innovative indications for the possible human use, even if, in my opinion, the use as functional food should be more investigated.

Response: Thank you for your comment. According to your comment, the more information about the use of rapeseed meal has been added in Line 28-30, 33-34 and 36-40 of revised manuscript. According to your comment, some innovative indications for the possible human use has been presented in conclusion (Line 737-762 of the revised manuscript, and it has been highlighted in yellow): one is that the removal of these anti-nutritional factors (glucosinolate, phytates and so on) make the rapeseed protein available in food applications (Line 746-747 of the revised manuscript, and it has been highlighted in green); another one is that the functionality of the FRSMP has been evaluated by in vitro and in vivo experiment, to develop the commercial products in food industry, the safety, gastrointestinal stability and potential sensory problems will be further investigated by the human clinical trial (Line 755-760 of revised manuscript, and it has been highlighted in green). Another innovative indications for the possible human use is that the fermentation products with high activity will be prepared into nano particle or microcapsules to apply as food additives and functional ingredients, which provides a thinking for the application of rapeseed meal in the possible human use.

The paper is described in details, with chemical and biological analyses. The data are well discussed to confirm the initial proposal of a combined microrganisms to produce small peptide with good activity. The subdivision into paragraphs helps the reading and focuses attention on the subject of the paragraph.

Response: Thank you for your comment. According to your comment, the long  paragraphs with several themes have been divided into several paragraphs, which helps the reading and focuses attention on the subject of the paragraph. The first paragraph of 4.1.2. (Discussion on the optimization of fermentation conditions) has been divided into three paragraphs, and the second paragraph of 4.1.2. have been divided into two paragraphs. The second paragraph of 4.3 (Discussion on the change of the molecular weight distribution and free amino acid composition of the RSM after fermentation) has been divided into two paragraphs.

If I can give a suggestion to the authors, in the conclusions I would better specify the possible use of the fermented product (how it can be used?? in which formulation it could be proposed???), to give more prominence to the possible human use, since many literature involves the use of fermented rapeseed for animal use.

Response: Thank you for your comment. According to your comment, the possible application of the fermented product and the prominence to the possible human use have been added in the conclusion (Line 746-762 of revised manuscript).

Many literature involves the use of fermented rapeseed for animal use, which was due to the presence of these anti-nutritional factors (glucosinolate, phytates and so on), thereby limiting the application of rapeseed meal in food field. In deed, the removal of these anti-nutritional factors (glucosinolate, phytates and so on) make the rapeseed protein apply to the food field available. In future, the fermentation products with high activity might be prepared into the nano particle or microcapsules to apply as food additives or functional ingredients.

Although the functionality of the FRSMP has been evaluated by in vitro and in vivo experiment, however, the application is still in the stage of laboratory research, which presents a certain gap from the commercial products in market application. To develop the commercial products in food industry, the safety, gastrointestinal stability and potential sensory problems will be further investigated by the human clinical trial.

The number of references is good and with recent date. Only the references n. 28-29-38- and 58 miss the year written in bold.

Response: Thank you for your comment. According to your comment, the year of  the references n. 28-29-38- and 58 have been written in bold in the reference 31  (Line 877-880 of revised manuscript), 32 (Line 881-883 of revised manuscript), 43 (Line 916-919 of revised manuscript) and 66 (Line 998-1001 of revised manuscript) of revised manuscript.  

The revised manuscript has been uploaded in the  attachment. Please see the attachment. Thank you very much!

Round 2

Reviewer 1 Report

I am satisfied with revised version of manuscript.

Author Response

Dear reviewer:

Thank you very very much for your help! We are very appreciated to receive your positive reply. Thanks a million. We are truly grateful for your comments, which improves the quality of the manuscript.  

According your comment, the English has been again improved by the native English teacher, and the spellings and grammars have been checked carefully. Any language revisions made to the manuscript have been marked using the “Track Changes” function of the MS Word/LaTeX, such that these changes can be easily reviewed by you. All the changes have been highlighted in red and described as follows.

We truly hope we can receive the positive reply again. If there is any question, you feel free to contact us, and we are very willing to improve our manuscript at all times until you are satisfied. Thank you very very much! Everything goes well! Best regards for you. 

Sincerely yours,

Yan Wang

The changes are as follows:

  1. The word “substilis”in title has been changed to “subtilis”, and the word “substilis” Line 117, 118, 265, 266, 472, 474, 505, 526 of revised manuscript (previous edition) have been changed to the word “subtilis” in Line 119, 120, 273, 274, 482, 484, 515 and 536 of revised manuscript with the track changes.
  2. The word group “well balanced”in Line 28 of revised manuscript (previous edition) has been changed to “well-balanced” in Line 28 of revised manuscript with the track changes.
  3. The word “l”of “which imites” has been changed to “which limits” in Line 33 of revised manuscript with the track changes.
  4. The sentence “are promising synergistic effects”in Line 59-60 of revised manuscript (previous edition) has been changed to the sentence “promisingly produce synergistic effects” in Line 62 of revised manuscript with the track changes.
  5. The “optimized strain”in Line 95 of revised manuscript (previous edition) has been changed to “an optimized strain” in Line 97 of revised manuscript with the track changes.
  6. The “ingredient contributed”in Line 99 of revised manuscript (previous edition) has been changed to “ingredient that contributes” in Line 101 of revised manuscript with the track changes.
  7. The “5KDa”in Line 187 of revised manuscript (previous edition) has been changed to “9.5 kDa” in Line 195 of revised manuscript with the track changes.
  8. The “an critical”in Line 354 of revised manuscript (previous edition) has been changed to “a critical” in Line 367 of revised manuscript with the track changes.
  9. The “processed with with”in Line 364-365 of revised manuscript (previous edition) has been changed to “processed with” in Line 377 of revised manuscript with the track changes.
  10. The blanks before and after “p”in Line 403 of revised manuscript (previous edition) have been deleted, and presented in Line 414 of revised manuscript with the track changes.
  11. The blanks before and after “pin Line 422 of revised manuscript (previous edition) have been deleted, and presented in Line 433 of revised manuscript with the track changes.
  12. The blank after Figure 7 in Line 427 of revised manuscript (previous edition) has been deleted, and presented in Line 438 of revised manuscript with the track changes.
  13. The “reported showed”in Line 506 of revised manuscript (previous edition) has been changed to “reported” in Line 516 of revised manuscript with the track changes.
  14. The blanks before and after “<”in “MW < 10 kDa” in Line 522 of revised manuscript (previous edition) have been deleted, and changed to “MW<10 kDa” in Line 532 of revised manuscript with the track changes.
  15. The “drastically reduction”in Line 543 of revised manuscript (previous edition) has been changed to “drastical reduction” in Line 553 of revised manuscript with the track changes.
  16. The “that”after “because” in Line 569 of revised manuscript (previous edition) has been deleted, and presented in Line 579 of revised manuscript with the track changes.
  17. The “flaven-3ols”in Line 589 of revised manuscript (previous edition) has been changed to “flavan-3ols” in Line 599 of revised manuscript with the track changes.
  18. The “3000 Da”in Line 611 of revised manuscript (previous edition) has been changed to “3 kDa” in Line 621 of revised manuscript with the track changes.
  19. The “of free EAA increased of FRSMP”in Line 623 of revised manuscript (previous edition) has been changed to “of free EAA of the FRSMP increased” in Line 633 of revised manuscript with the track changes.
  20. The “,”in Line 639 of revised manuscript (previous edition) has been changed to “: ” in Line 649 of revised manuscript with the track changes.
  21. The “short chained”in Line 641 of revised manuscript (previous edition) have been changed to “short-chained” in Line 651 of revised manuscript with the track changes.
  22. The “methanism”in Line 665 of revised manuscript (previous edition) has been changed to “mechanism” in Line 675 of revised manuscript with the track changes.
  23. The “the the”in Line 728 of revised manuscript (previous edition) has been deleted in Line 738 of revised manuscript with the track changes.
  24. The “however”in Line 756 of revised manuscript (previous edition) has been deleted.
  25. The “in market application”in Line 757 of revised manuscript with previous edition has been changed to “in the market” in Line 767 of revised manuscript with the track changes.

Reviewer 2 Report

Dear Author, I reviewed the revised version of the manuscript (foods-1778642) entitled: A novel fermented rapeseed meal, inoculated with selected protease assisting screened B.substilis YY-4 and L.plantarum 6026, showed high availability, strong antioxidant and immunomodulation potential capacity. This version of the manuscript followed all the suggested modifications and recommendations by the reviewers. However, try to include figure descriptions for Figures 1 and 2. Besides, the findings obtained in this research are well described and compared with bibliographical references and justify the data's importance. For this reason, I consider that this manuscript can be accepted for publication in this journal.

Author Response

Dear reviewer:

  We are very very appreciated to receive your positive reply. Thank you very very much for your help! We are truly grateful for your comments, which improves the quality of the manuscript. Thank you again! According your comment, the figure descriptions for Figures 1 and 2 have been added in Line 45-46 and Line 169-174 of revised manuscript with the track changes, and all the changes have been highlighted in red. The spellings and grammars of the manuscript have been again checked and improved by a native English teacher.

  We sincerely hope that we receive the positive reply again for the figure descriptions. If there is any question, you feel free to contact us, and we are very willing to improve our manuscript at all times until you are satisfied. Thanks a million! Everything goes well. Best regards for you. 

Sincerely yours,

Yan Wang

Reviewer 3 Report

Thanks to the authors for making the required changes. 

Author Response

Dear reviewer:

  Thank you very very much for your help! We are very appreciated to receive your positive reply. We are truly grateful for your critical comments for improving the quality of the manuscript. Thanks a million.

  Everything goes well. Best regards for you. 

Sincerely yours,

Yan Wang
